

# Classification of Arctic, Mid-Latitude and Tropical Clouds in the Mixed-Phase Temperature Regime

Anja Costa[1], Jessica Meyer[1,2], Armin Afchine[1], Anna Luebke[1,3], Gebhard Günther[1], James R. Dorsey[4], Martin W. Gallagher[4], Andre Ehrlich[5], Manfred Wendisch[5], Darrel Baumgardner[6], Heike Wex[7], and Martina Krämer[1]

[1]Forschungszentrum Jülich GmbH, Jülich, Germany
[2]now at: Bundesanstalt für Arbeitsschutz und Arbeitsmedizin, Dortmund, Germany
[3]now at: Max Planck Institute for Meteorology, Atmosphere in the Earth System Department, Hamburg, Germany
[4]Centre for Atmospheric Science, University of Manchester, UK
[5]Leipziger Institut für Meteorologie, Universität Leipzig, Germany
[6]DMT, Boulder/Colorado, USA
[7]Leibniz Institute for Tropospheric Research, Leipzig, Germany

*Correspondence to:* Martina Krämer (m.kraemer@fz-juelich.de)

**Abstract.** The degree of glaciation of mixed-phase clouds constitutes one of the largest uncertainties in climate prediction. In order to better understand cloud glaciation, cloud spectrometer observations are presented in this paper that were made in the mixed-phase temperature regime between 0°C and -38$^o$C, where cloud particles can either be frozen or liquid. The extensive dataset covers four airborne field campaigns providing a total of 139,000

1 Hz data points (38.6 hours within clouds) over Arctic, mid-latitude and tropical regions. We develop algorithms combining the information on number concentration, size and asphericity of the observed cloud particles to classify four cloud types associated with liquid clouds, clouds where liquid droplets and ice crystals coexist, fully glaciated clouds after the Wegener-Bergeron-Findeisen process, and clouds where secondary ice formation occurred. We quantify the occurrence of these cloud groups depending on the geographical region and temperature and find that

liquid clouds dominate in our measurements during the Arctic spring, while clouds dominated by the Wegener-Bergeron-Findeisen process are most common in mid-latitude spring. Coexistence of liquid water and ice crystals is found over the whole mixed-phase temperature range in tropical convective towers in the dry season. Secondary ice is found at mid-latitudes at -5 °C to -10 °C and at higher altitudes, i.e. lower temperatures in the tropics. The distribution of the cloud types with decreasing temperatures is shown to be consistent with the theory of evolution

of mixed-phase clouds. With this study, we aim to contribute to a large statistical database on cloud types in the mixed-phase temperature regime.

## 1 Introduction

Clouds can be classified according to their altitude (low, mid-level, high, see e.g. Rossow and Schiffer, 1991), their temperature (warm, cold) or their cloud particle phase (liquid, mixed-phase: both liquid and ice, ice). Especially





for intermediate altitudes, these classification criteria overlap: ice particles may sediment into warm cloud layers; updrafts can transport liquid water droplets into colder cloud regions; droplet formation may produce liquid water content in a cold, formerly glaciated cloud (Findeisen et al., 2015; Korolev, 2007).

To avoid ambiguities, we refer here to all clouds observed at temperatures between 0 °C and -38 °C as 'clouds
in the mixed-phase temperature regime' (mpt clouds). In that temperature regime, purely liquid (supercooled) clouds can be found as well as mixed-phase clouds (where liquid water droplets and ice crystals coexist) and also fully glaciated clouds (Pruppacher et al., 1998). Within this temperature range, important processes take place that transform the cloud's phase or microphysical characteristics significantly. This phase transition is not only an important part of precipitation-forming processes like the cold rain process, it also affects the cloud's radiative
properties by influencing the solar albedo of mpt clouds in the sense that with growing ice fraction, their solar albedo (cooling) effect is reduced (Ehrlich et al., 2009; Wendisch et al., 2013). Thus a correct representation of this cloud type in global climate models is of importance for an improved certainty of climate predictions (Wendisch et al., 2013).

The transformation from a fully liquid to a fully frozen cloud can follow different, sometimes non-linear paths, as
illustrated in Figure 1. After the activation of cloud condensation nuclei formed small droplets < 50 $\mu$m (all-liquid state), initial freezing can occur in those droplets that contain an ice nucleating particle (INP) that can be activated in the ambient cloud environment (resulting in a mixed phase state: coexistence of ice and water). Different INP can induce ice nucleation at different temperatures, depending on their nature, e.g. if they are of biological or mineral origin, their morphology, and freezing efficiency. Therefore, the number of droplets containing an INP
to heterogeneously form ice is important for its glaciation, and also the temperature of the mpt cloud is relevant, as the freezing efficiency of different INP varies with temperature. The nature of the INP properties that favor ice formation is one of the major open questions in cloud and climate research. This is summarized in the recent review article by Kanji et al. (2017) and references therein. The conditions that favor drop freezing are - in a simplified summary: cold temperatures, high relative humidities and a 'good freezing ability'. Biological particles are known
to induce ice nucleation in the temperature range between about 0 to -20 °C, while mineral dust particles initiate ice at temperatures below about -20 °C (Kanji et al., 2017; Augustin-Bauditz et al., 2014).

The persistence of supercooled liquid clouds in case no ice active INP are present is also reported by Korolev (2007). Moreover, the further development of the glaciation degree of a mpt cloud, where a few ice crystals are present, is discussed in this study with dependence on the environmental dynamical conditions. This is illustrated
by theoretical considerations (Korolev, 2007) of the partitioning of liquid and ice water content in rising mixed-phase cloud parcels under different conditions (see Figure 2, adapted from Korolev, 2007, with modifications). The first scenario represents an intermediate vertical velocity (1 m s$^{-1}$; blue lines), where the Wegener-Bergeron-Findeisen process (Findeisen et al., 2015) is triggered above the altitude marked by the blue line (note that the temperature decreases with increasing altitude), which leads to full glaciation of the cloud. At that point, the
relative humidity over water falls below 100% (RHw < 100%), as more and more water vapour is consumed by the





many small liquid cloud droplets. As a result, these droplets evaporate, decreasing the liquid water content. The RH over ice remains above 100% (RHi > 100%), allowing the few ice crystals to grow to large sizes > 50 $\mu$m, thus increasing the ice water content.

In contrast, the red graphs show a scenario for higher vertical velocities (2 m s$^{-1}$). Here, due to the high up-
draft, the supersaturation is preserved over both water and ice (RHw, RHi > 100%) over the complete altitude range. Subsequently, the liquid and ice water content increase in coexistence and the cloud continues to be only partly glaciated ('coexistence cloud'). These simulations demonstrate that vertical velocity is a major parameter controlling the occurrence of different cloud types, because the updraft is the crucial parameter for possible supersaturations. The supersaturation over water can remain at or above 100% only in high updrafts, thus allowing
coexistence clouds to survive down to about -38 °C, where the supercooled liquid cloud droplets will freeze homogeneously (Pruppacher et al., 1998; Koop et al., 2000). Also, secondary ice production can take place producing high number concentrations of small ice particles (see overview in Field et al., 2015, 2017). Known processes are e.g. the Hallett-Mossop process (Hallett and Mossop, 1974) and ice-ice collisions (Yano and Phillips, 2011). When one of these processes has started, the remaining liquid fraction of a cloud can glaciate quickly via contact
freezing.

Evaporation of both numerous small liquid droplets and large ice particles occurs when the environment is subsaturated with respect to both water and ice (RHi < 100%, RHw < 100%), as predicted by Korolev (2007) for downdraft regions within the cloud. If this state persists sufficiently long, the cloud will fully evaporate.

In summary, as illustrated in Figure 1 and Table 1, four types of mpt clouds are expected to occur: The first type
describes purely liquid clouds with many small (diameter < 50 $\mu$m) liquid droplets that appear often at slightly supercooled conditions and with lesser frequencies as the temperature becomes colder (Bühl et al., 2013). This cloud type may additionally contain a low concentration of large particles (large droplets from coalescence or ice particles sedimenting from above). The second cloud type are coexistence clouds with a high concentration of small cloud particles < 50 $\mu$m that can be liquid or frozen. The coexistence cloud type appears at decreasing temperatures
in higher updrafts. In case the updrafts are very strong as in tropical convective clouds, the supercooled liquid cloud droplets can reach cold temperature regions around -38 °C and freeze homogeneously. Furthermore a third type with a high concentration of small ice particles (diameter < 50 $\mu$m) might emerge as a result of secondary ice production e.g. due to the Hallett-Mossop process at temperatures between -3 °C and -8 °C or ice splintering. A fourth cloud type in case of lower updrafts, fully glaciated Wegener-Bergeron-Findeisen (WBF) clouds containing
only very few or no small liquid droplets (< 50 $\mu$m), but consisting mostly of large ice crystals, are expected to appear with increasing frequency when the temperature decreases.

Due to the manifold interactions between large-scale and small-scale dynamics, aerosol/INP availability, and complex processes of formation and evolution of supercooled liquid and frozen cloud particles, mpt clouds are not well understood and therefore poorly represented in global climate models (Boucher et al., 2013). As a conse-
quence, the uncertainties concerning the global mpt cloud cover's radiative impact are large. Of particular interest





is the partitioning of ice and liquid water, i.e. the glaciation degree. An important step to improve the incomplete understanding of the phase transition processes are reliable observations of the different types of mpt clouds. However, cloud particle phase observations are limited by technical constraints: passive satellite data mostly provide information on cloud tops, ground-based lidars can not quantify thick layers of liquid water (Shupe et al., 2008;

Storelvmo and Tan, 2015). Active sensors have been used to derive liquid and ice water paths for the full depth of the atmosphere (reported in Boucher et al., 2013, p.580), but are subject to large errors. In situ measurements may cover the full vertical extent (Taylor et al., 2016; Lloyd et al., 2015; Klingebiel et al., 2015), but are restricted to the flightpath and have to be analysed carefully (Wendisch and Brenguier, 2013). Also, the phase identification often depends on cloud particle sizes. Small cloud particles $< 50\,\mu$m are usually regarded as liquid (see e.g. Taylor

et al., 2016). With particle imaging probes like OAPs (Optical Array Probes), more sophisticated shape recognition algorithms can be used (e.g. Korolev and Sussman, 2000), which are nevertheless limited. Usually, they require a minimum number of pixels (corresponding to cloud particles with diameters of $70\,\mu$m and more) to recognize round or aspherical particles reliably.

In this paper, we use in situ airborne cloud measurements in the cloud particle size range from $3\,\mu$m to $937\,\mu$m

to classify the above described types of clouds in the mpt regime (see Figure 1): 'Mostly liquid' clouds after drop formation, 'coexistence clouds' after initial freezing, 'secondary ice' clouds influenced by ice multiplication, and clouds after the WBF process. This classification enables us to revisit a statistical overview published by Pruppacher et al. (1998), stating at which temperatures purely liquid or ice-containing clouds were found.

For all except the fourth cloud type, we expect high cloud particle number concentrations with a peak at

20 cloud particle sizes $< 50\,\mu$m. Thus, particle size distributions and concentrations allow the differentiation between glaciated clouds mainly formed via the WBF process and other cloud types in the mpt regime. To identify these other types more closely, we use a new detector that can measure the asphericity of the small ($< 50\,\mu$m) cloud particles (Baumgardner et al., 2014) together with a visual shape inspection of particles $> 50\,\mu$m. The occurrence of the four cloud types is quantified with regard to measurement location and temperature by performing a statistical

analysis of the 1 Hz data.

The article is structured as follows: in section 2, the field campaigns are described, as well as the cloud spectrometer NIXE-CAPS and its data products. In section 3, the observations are evaluated with respect to the clouds' size distribution, the correlation of cloud particle concentrations to expected ice nucleating particle concentrations, the cloud particle asphericity and the associated vertical velocities. section 4 summarizes the findings of this study.

## 2   Methodology

Four airborne field campaigns were performed in Arctic, mid-latitude and tropical regions (see subsection 2.1). In total, the dataset in the mixed-phase temperature regime between $0\,°$C and $-38\,°$C covers 38.6 hours. Mpt clouds





were measured using the cloud spectrometer NIXE-CAPS (see subsection 2.2). The data analysis is described in subsection 2.3.

## 2.1 Field campaigns

The first campaign, COALESC (Combined Observation of the Atmospheric boundary Layer to study the Evolution of StratoCumulus), was based in Exeter, UK, in February and March 2011. The NIXE-CAPS was installed as a wing probe on the BAe146 aircraft operated by the Facility for Airborne Atmospheric Measurements (FAAM), UK. All flights took place in the coastal area of south-east England and Wales; the main campaign targets were low stratus and stratocumulus clouds. The campaign is described in Osborne et al. (2014), Table 2 provides an overview of the flights. Out of 16 measurement flights, 14 provided observations of mpt clouds, with in total 41042 seconds (11.4 hours) of data.

Measurements in Arctic clouds have been conducted during the campaigns VERDI (April and May 2012, Study on the VERtical Distribution of Ice in Arctic clouds, see also Klingebiel et al., 2015) and RACEPAC (April and May 2014, Radiation-Aerosol-Cloud ExPeriment in the Arctic Circle). Both campaigns took place in Inuvik, Northern Canada. Research flights were performed with the Polar-5 and Polar-6 aircraft of the Alfred-Wegener-Institut, Germany. The 13 flights of both VERDI (see Table 3) and RACEPAC (Table 4) covered the region of the Arctic Beaufort Sea coast with its retreating sea ice in spring. VERDI yielded 59028 seconds (16.4 hours) of observations within mpt clouds, RACEPAC contributed 33354 seconds (9.3 hours). Although both campaigns took place at the same time of the year, different synoptic situations lead to different cloud characteristics: VERDI was dominated by stable anticyclonic periods with weak gradients of atmospheric parameters that allow the formation of a strong inversion in the boundary layer associated with persisting stratus, whereas during RACEPAC frontal systems frequently passed the area of the observations and lead to a more variable and short-lived cloud situation.

The tropical measurement campaign ACRIDICON-CHUVA (Aerosol, Cloud, Precipitation, and Radiation Interactions and Dynamics of Convective Cloud Systems/Cloud processes of tHe main precipitation systems in Brazil: A contribUtion to cloud resolVing modelling and to the GPM (Global Precipitation Measurements)) was carried out in September and October 2014. The instrument platform was HALO (High Altitude and Long Range Research Aircraft), a Gulfstream V aircraft operated by DLR (Deutsches Luft- und Raumfahrtszentrum/German Aerospace Centre). Based in Manaus, Brazil, ACRIDICON-CHUVA was aimed at convective clouds over tropical rainforest and deforested areas (cf. Table 5; for details, see Wendisch et al., 2016). The campaign comprises 14 flights, 11 of which contained clouds in the mixed-phase temperature regime. Although cloud profiling at various altitudes and temperatures was a main directive of ACRIDICON-CHUVA, the total time spent within mpt clouds was only 5368 seconds (1.5 hours). The relatively limited time span was caused by the high flying speed of HALO (up to $240 \, \mathrm{m \, s^{-1}}$); it results in short penetration times (in the range of several seconds) of the convective towers. A second reason is the increasing danger of strong vertical winds and icing in developing cumulonimbus clouds. From





certain cloud development stages on, only the cloud's anvil and outflow at cold temperatures lower than -38 °C could be probed.

## 2.2 The NIXE-CAPS instrument

The observations presented here comprise particle number concentrations, size distributions and shape information obtained by NIXE-CAPS (New Ice eXpEriment: Cloud and Aerosol Particle Spectrometer). Two instruments are incorporated in NIXE-CAPS (Baumgardner et al., 2001; Meyer, 2012; Luebke et al., 2016): the NIXE-CAS-DPOL (Cloud and Aerosol Spectrometer with Detection of POLarization) and the NIXE-CIPg (Cloud Imaging Probe - Greyscale). In combination, particles with diameters between $0.61\,\mu$m and $937\,\mu$m can be sized and counted; NIXE-CAPS measurements are thus split into an aerosol dataset (particle diameters $0.61\,\mu$m to $3\,\mu$m) and cloud particle dataset (particle diameters $3\,\mu$m to $937\,\mu$m). For aircraft speeds between $240\,\mathrm{m\,s^{-1}}$ and $80\,\mathrm{m\,s^{-1}}$, the instruments' sampling volumes limit the particle concentration measurements to concentrations above $0.02\,\mathrm{cm^{-3}}$ to $0.05\,\mathrm{cm^{-3}}$ (NIXE-CAS-DPOL) and about $0.0001\,\mathrm{cm^{-3}}$ to $0.001\,\mathrm{cm^{-3}}$ (NIXE-CIPg; the exact values depend on the particle size, see Knollenberg, 1970)). The instrument is mounted below the aircraft wing. A detailed description of the operating principles, limitations and uncertainties can be found in Meyer (2012). The overall measurement uncertainties concerning particle concentrations and sizes are estimated to be approximately 20% (Meyer, 2012).

As an improvement over former instrument versions, NIXE-CAPS was modified to minimize ice crystal shattering on the instrument housing, because those ice fragments can artificially enlarge the ice particle concentrations (Field et al., 2006; Korolev and Field, 2015). Therefore, the tube inlet of the NIXE-CAS-DPOL has been sharpened to knife-edge, and K-tips have been attached to the NIXE-CIPg's arms (Korolev et al., 2013; Luebke et al., 2016).

In the following, we present an overview of the two instrument components NIXE-CAS-DPOL and NIXE-CIPg as well as the data analysis.

### 2.2.1 NIXE-CAS-DPOL - particle asphericity detection

The NIXE-CAS-DPOL (hereafter referred to as the CAS) covers the small particle size range between $0.61\,\mu$m and $50\,\mu$m. As particles pass through the spectrometer's laser beam, the forward-scattered light intensity is used for particle sizing (Baumgardner et al., 2001). As a new feature, the CAS records the change of polarization in the backward scattered light, thus giving information about the particle asphericity (Baumgardner et al., 2014). Light scattered by spherical particles in the near-backward direction (168°-176°) will retain the same angle of polarization as the incident light. In contrast, depending on the amount of asphericity, light scattered by non-spherical particles will have some components that are not at the same incident light polarization. The CAS uses a linearly polarized laser and two detectors that measure the backscattered light. One detector is configured to only detect scattered light with polarization that is perpendicular (cross-polarized) to the incident light. This signal is referred to as S-pol. In Figure 4, we show that the intensity of the S-pol signal generates characteristic values for both spher-





ical and aspherical particles. The signature of spherical particles is measured in warm cloud sections (T > 0 °C), if possible during each measurement campaign. Figure 4 shows an example obtained during the ACRIDICON-CHUVA campaign: measurements of the cross-polarized light as a function of cloud particle size are shown for both a liquid and a glaciated cloud. The liquid spherical particles cause only a very weak S-pol signal. From this

measurement, we derive an asphericity threshold (see black line in Figure 4), providing a method to distinguish between spherical and aspherical particles. This asphericity threshold is verified, if possible, during each of the airborne campaigns by analyzing a flight segment in clouds warmer than 0 °C. The S-pol signal caused by ice particles is shown in Figure 4 (right panel) for a cirrus cloud (at -60 °C). Clearly, the ice crystals cause strong S-Pol signals above the asphericity threshold. It can also be seen that the signal strength depends on the size of the crys-

tals. In particular, the instrument sensitivity with regard to particle asphericity decreases for particles smaller than 20 $\mu$m (note that the particles with diameters smaller than 3 $\mu$m are aerosol particles). This was found during the experiments described by Järvinen et al. (2016), who compared several asphericity detection methods, including the CAS. Järvinen et al. (2016) also show that ice crystals can be near spherical. The low signal caused in the CAS polarisation detector by this type of crystals can lead to an underestimation of the glaciation degree of a mixed

phase cloud if it is derived from aspherical cloud particle fractions (see also Nichman et al., 2016). In addition, there are variations in the S-pol signals that are caused by the orientation of the crystal with respect to the laser beam (Baumgardner et al., 2014).

    Taking into account these uncertainties, we find that it is possible to use the S-pol signal for a classification of mpt clouds. Firstly, we perform the asphericity analysis only for particle sizes between 20 $\mu$m and 50 $\mu$m, the range with

the strongest S-pol signal. For this size range, we derive 'aspherical fractions' (AF): the percentage of aspherical particles per second, which means that particle bulk properties are analyzed, not single particle signatures alone. Secondly, we do not interpret each aspherical fraction measurement alone, but divide the AFs into three groups: (i) AF = 0% (zero), (ii) AF: 0 - 50% (low) and (iii) AF = 50 - 100% (high).

### 2.2.2 NIXE-CIPg

The NIXE-CIPg (called CIP from here on) is an optical array probe (OAP) that nominally records particles between 7.5 $\mu$m and 960 $\mu$m by recording images with shadow intensities of 100%-65%, 65-35%, and 35%-0% of the incident light. Particle sizes and concentrations are derived by using the SODA2 program (Software for OAP Data Analysis, provided by A. Bansemer, National Center for Atmospheric Research NCAR/University Corporation for Atmospheric Research UCAR, 2013). For a detailed description of SODA2, see for example Frey (2011). Pixels

with shadow intensities of 35% and higher were used for the image analysis. In the observations presented here, only the number concentrations for particles with diameters > 22 $\mu$m are taken from the CIP dataset. The smaller particle fraction is covered by the CAS measurements. The shadow images can be analysed for particle asphericity using various algorithms (Korolev and Sussman, 2000); in this study, however, the occurrence of irregular (i.e. ice) particles was verified manually.



## 2.3 Data analysis

NIXE-CAPS records four individual datasets: 'histogram' and 'particle by particle' (PBP) data for both the CIP and the CAS instrument. All datasets are evaluated using the NIXElib library (Meyer, 2012; Luebke et al., 2016). In the 1 Hz histogram datasets, particles are sorted into size bins according to predefined forward scattering cross sections (CAS) or maximum shadow diameters (CIP). With these, histograms are created for every second.

The PBP dataset recorded by the CIP consists of a time stamp and the shadow image of each individual particle. The shadow images can be analysed with regard to maximum diameter, equivalent size, area ratio, and shape. The CAS PBP data are limited to 300 particles per second. For these particles, detailed information is stored: the forward, backward P-pol and backward S-pol scattering intensities, a time stamp, and the particle inter-arrival time.

Apart from the asphericity analysis, this dataset also allows a diagnosis of ice crystal shattering following Field et al. (2006) and Korolev and Field (2015). Thus, an inter-arrival time (IAT) correction was applied (Field et al., 2006) additionally to the instrument modifications described above. IAT histograms compiled during the data analysis showed only very few measurements with short IATs, during which a maximum of about 5% of the cloud particle population might result from shattering.

## 3 Results and Discussion

### 3.1 Mpt cloud classification based on particle number size distributions

Four cloud types are expected in the mpt regime (see Table 1). As mentioned in the introduction, however, only two typical particle number size distributions (PSD) are found frequently in mpt clouds. Figure 5 shows NIXE-CAPS PSDs measured during VERDI flight 08, where both types alternate: some cloud regions show very high particle concentrations of small particles with a mode diameter $< 50\,\mu$m (see example PSD in the lower right corner). Alternatively, the clouds consist mostly of large ice crystals $> 50\,\mu$m with either no small particles or concentrations below the NIXE-CAS detection limit (see example PSD in the lower left corner).

As a first step of the mpt cloud classification, we sort all clouds according to their particle size distribution type and address these types separately. To this end, we calculate two cloud particle number concentrations, one for particles with diameters between $3\,\mu$m and $50\,\mu$m ($N_{small}$) and one for all larger particles ($N_{large}$). For the classification of the first cloud type (Type 1), $N_{small}$ must exceed $1\,cm^{-3}$, while $N_{large}$ can be zero or larger. The mode of the cloud particle mass distribution is at particle diameters $< 50\,\mu$m. We assume that this type matches the young clouds after droplet condensational growth in Figure 1. In the second cloud type (Type 2) we classify those clouds with $N_{small}$ below $1\,cm^{-3}$ and $N_{large}$ present. The mode of the cloud particle mass distribution is here at particle diameters $> 100\,\mu$m. This type matches fully glaciated clouds, e.g. as a result of the WBF process (see Figure 1).





In Figure 6, a histogram is provided that shows the occurrence of cloud particle concentrations throughout our dataset. The spectrum of observed concentrations is continuous, but the two modes associated with the Type 1 and Type 2 clouds (as described above) are clearly visible. The area between the two modes (a total of 6% of all observations) might result from clouds in a 'transition' state to glaciation. In this study, these measurements were

assigned to Type 1 clouds. The smallest mode with a peak at around $10^{-4}\,\mathrm{cm}^{-3}$ shows concentrations around the detection limit of the CIP (a total of 5% of all observations). We assume that these are measurements in precipitation, especially in snow that occurred frequently in the Arctic campaigns, and in sedimenting aggregates of ice crystals from tropical convective clouds (see subsection 3.3).

In the following, we discuss the cloud types described above in more detail. Type 1 cloud characteristics mea-

sured during VERDI are shown in Figure 7). These clouds have a clear mode between $3\,\mu\mathrm{m}$ and $50\,\mu\mathrm{m}$ and are very dense, cloud particle number concentrations reach average values of dozens to more than two hundred $\mathrm{cm}^{-3}$. Table 6 shows average cloud particle concentrations for the Type 1 clouds in 5 K intervals. Low number concentrations of large ice particles $> 50\,\mu\mathrm{m}$ are sometimes found, but all clouds of this type are dominated by $N_{\mathrm{small}}$, which may consist of liquid droplets, frozen droplets, or small ice from ice multiplication processes. With regard to the

concentrations of $N_{\mathrm{small}}$ in the different temperature intervals (Figure 7 and Table 6), it can be clearly seen that they decrease with decreasing temperature. When a cloud consists of liquid droplets, they grow by condensation when lifted to higher altitudes - and thus colder temperatures - followed by an increasing coalescence of the droplets, which consequently causes a higher number of $N_{\mathrm{large}}$ while depleting the concentration of small droplets. This is also visible in Figure 7. Note, however, that $N_{\mathrm{large}}$ also decreases with increasing temperature, reaches a min-

imum around 260 K, and then rises again, possibly reflecting the increasing occurrence of sedimenting particles. Visual inspection of the CIP images indicates that in the $N_{\mathrm{large}}$ cloud mode ice crystals can be found in addition to the drizzle drops. Three of the cloud types of the mpt regime are expected to show Type 1 cloud characteristics: 'liquid', 'coexistence' and 'secondary ice' clouds.

The second set of PSDs (Type 2: Figure 8) is not strongly dominated by $N_{\mathrm{small}}$. Here, $N_{\mathrm{large}}$ form a distinct

mode. Both mode concentration and maximum values decrease with decreasing temperatures. Clouds of this PSD type have low number concentrations of - on average - less than $0.1\,\mathrm{cm}^{-3}$ in the size range 3 to $50\,\mu\mathrm{m}$ (see Table 6). For the sizes $> 50\,\mu\mathrm{m}$, the CIP images show ice crystals or aggregates. This is the typical appearance of a fully glaciated cloud, formed either via the WBF process during which the small liquid droplets evaporate or, at lower altitudes (higher temperatures), due to sedimentation, when aggregates precipitate from higher levels. Again, the

two temperature groups are seen as for the Type 1 clouds (Figure 7). An explanation can be that Type 2 clouds most probably develop from Type 1: once the environment becomes subsaturated (RHw < 100%, RHi > 100%), all liquid droplets evaporate leaving only the ice crystals that have already formed from droplets that contain an INP. Therefore, $N_{\mathrm{large}}$ of Type 2 is only a fraction of those of Type 1, which might reflect the number of active INP in the respective temperature interval in case no ice multiplication takes place (see subsection 3.2). Thus, the larger

differences between the two temperature groups - as seen for Type 1 clouds - more or less balance out. Indeed,




an increase of average ice crystal numbers can be seen (Table 6, bottom, $N_{large}$), which might be interpreted as increasing fraction of activated INP with decreasing temperature. Note that $N_{small}$ is still larger than $N_{large}$. Since shattering artifacts are unlikely (cf. subsection 2.3), this means that in Type 2 clouds, a significant number of small particles persists over the whole temperature range, too.

In addition to these two types, thin clouds with only low concentrations (less than $1\,cm^{-3}$) of small particles ($< 50\,\mu m$) and no large particles are sometimes found, which are most likely evaporating clouds. They are not considered as a separate cloud type, since they do not appear frequently and can not be regarded as a distinct type, they are remnants of one of the two cloud types defined above. Further, the respective measurements stem from the CAS instrument alone and are close to its detectable concentration limit, thus suffering from an enhanced
uncertainty.

### 3.2   Comparison of cloud particle with ice nucleating particle numbers

A comparison of the measured cloud particle number concentrations to INP concentrations ($N_{INP}$) can give an indication if the ice particles may result from primary ice nucleation. No direct INP measurements are available for our data set, so we estimated $N_{INP}$ using the formula provided by DeMott et al. (2010), where aerosol numbers
of particles between $0.5\,\mu m$ and $3\,\mu m$ are related to INP concentrations. NIXE-CAPS records particles larger than $0.6\,\mu m$; the fraction from $0.6\,\mu m$ to $3\,\mu m$ is used as 'aerosol fraction'. The results for $N_{INP}$ are shown in Figure 9 as a function of temperature. Generally, $N_{INP}$ increases with decreasing temperature, as already mentioned in the last section. The most frequent $N_{INP}$ range between the lowest calculated value of $10^{-4}\,cm^{-3}$ ($0.1\,L^{-1}$) and $\sim 10^{-3}\,cm^{-3}$ ($1\,L^{-1}$), while the maximum reaches up to $0.3\,cm^{-3}$ ($\sim 300\,L^{-1}$). In comparison to a compilation
of INP measurements presented recently by Kanji et al. (2017), the estimated range of INP numbers is shifted to somewhat smaller concentrations.

In Figure 10, $N_{small}$ and $N_{large}$ for both Type 1 and Type 2 clouds are now shown in the same way of presentation as before $N_{INP}$. In Type 1 clouds, especially for $N_{small}$ (upper left panel), we find concentrations between $2\,cm^{-3}$ and more than $200\,cm^{-3}$ down to temperatures of -20 °C, well exceeding all INP estimations in this temperature
range. But also for $N_{large}$ (upper right panel), the cloud particle concentrations exceed the expected $N_{INP}$ by several orders of magnitude. For colder temperatures, where the measured cloud particle number concentrations are lower, the estimated $N_{INP}$ are also mostly lower than the cloud particle concentrations. In general, we can exclude primary ice nucleation as origin for cloud particles in the Type 1 clouds.

The $N_{large}$ of Type 2 clouds (lower right panel) agree quite well with $N_{INP}$ for a wide range of temperatures.
However, in warm areas, the cloud particle concentrations can be higher - they might represent large ice crystals sedimenting from upper layers, as mentioned in Section 3.1. For the colder regions, the agreement is consistent with the assumption that the Type 2 clouds we observed were formed by the WBF process (see subsection 3.1) and that the initial ice crystals have likely formed around INP. $N_{small}$ is slightly increased in comparison with



$N_{INP}$. Again, it is possible that this is an effect of the CAS' limited detectable concentration range, as discussed in subsection 2.2.

### 3.3 Mpt cloud classification based on particle asphericity

Size distributions, cloud particle number concentrations and comparisons with expected INP number concentrations provide little information on the cloud particle phase (cf. subsection 3.1, subsection 3.2). For further insights on the nature of the observed clouds, information on cloud particle asphericity is used.

As described in subsubsection 2.2.1, for $N_{small}$ we define three groups with regard to AFs (1 Hz data of 'aspherical fractions') to help classifying the mpt clouds: (i) AF = 0% (zero), (ii) AF: 0% to 50% (low) and (iii) AF: 50% to 100% (high).

Figure 11 shows the aspherical fractions of Type 1 and 2 cloud particles vs. temperature, the data points are color coded by the respective field campaigns. The horizontal lines show the 0 °C (liquid) and -38 °C (ice) temperature thresholds. Looking at the data points in pure ice clouds below -38 °C it can be seen that most of the measurements are found in group (iii) 'high AF' range. These AF can therefore be associated with fully glaciated clouds. Note that Type 2 clouds show AF comparable to those of cirrus clouds. The small particles found in relatively large number concentrations in this cloud type (see subsection 3.1 and Table 6) must therefore be small ice crystals.

Due to the ambiguities of the polarization measurement discussed in Section 2.2.1, AF covers a broad range, most often between 70%-80%. Note that even in the cirrus clouds the AF never reaches 100%. A possible reason for this deviation can be columnar ice crystals: these are not well recognized by the CAS sensor (see Järvinen et al., 2016). Alternatively, frozen droplets might have maintained their compact, quasi-spherical shape. All aspherical fractions derived from CAS measurements must therefore be seen as 'minimum aspherical fractions'.

AF found in group (i) - 'zero AF' - are classified as liquid, while AF observations in group (ii) - 'low AF' - are regarded as mixed-phase clouds (liquid + ice). Group (iii) - high AF - is most likely fully frozen. Particles > 100 $\mu$m are mostly irregular (i.e. ice) in group (ii) and (iii). In group (i), large ice particles can occasionally be found. In the size range between 50 $\mu$m and 100 $\mu$m, the distinction between drizzle drops and ice particles is not possible, because the shadow images do not contain enough pixels to differentiate between spherical and irregular particles (see subsection 2.2).

### 3.4 Cloud type detection in the mpt regime

The different cloud types that can be expected in the mpt regime (Table 1) can be identified by the combination of information about $N_{small}$, $N_{large}$, and the respective aspherical fractions (AF) in each size range. Following this line, we developed algorithms to sort the mpt clouds - second by second - into the four cloud types, using the following criteria:

1. 'Mostly liquid' Type 1 clouds are classified where $N_{small}$ is > 1 cm$^{-3}$ and AF is zero (liquid).





2. 'Coexistence' Type 1 clouds are classified where $N_{small}$ is $> 1\,cm^{-3}$ and AF is low ($< 50\%$, liquid and ice) and large ice crystals $N_{large}$ are present.

3. 'Secondary ice' Type 1 clouds are classified where $N_{small}$ is $> 1\,cm^{-3}$, AF is high (ice) and large ice crystals $N_{large}$ are present.

4. 'WBF/Large ice' Type 2 clouds are classified where $N_{small}$ is $< 1\,cm^{-3}$, AF is high ($> 50\%$, ice) and large ice crystals $N_{large}$ are present.

### 3.5 Mpt cloud classification: Results

The mpt clouds observed in this study were probed under a wide range of meteorological conditions (see subsection 2.1). We can therefore assume that these clouds have formed and evolved in different environments with regard to INP properties and updrafts, which are shown in the previous section to be the major parameters influencing the mpt cloud glaciation process.

For a comprehensive interpretation of the observed clouds, we divided the clouds into Artic, mid-latitude, and tropical clouds, analysed the vertical velocities from the aircraft's meteorological data for the different cloud types (Figure 12), estimated INP numbers (Figure 13) and finally established distributions of the four mpt cloud categories (see Section 3.4) as a function of temperature (note that the temperatures are related to different altitudes depending on the geographical region, Figure 14). The results are presented in Figure 15.

#### 3.5.1 Arctic clouds

The cloud types found during the field campaigns VERDI and RACEPAC are shown in Figure 15 (left panel). For the probed temperature ranges (253 to 273 K - note that the temperature values in the figure indicate midbins), 50 to 80% of the mpt clouds belong to the 'Mostly liquid' (pink) category. Further, we find a low number of 'Coexistence' clouds (brown) and a small percentage of glaciated 'WBF' clouds (dark blue). A possible explanation for the large amount of 'Mostly liquid' clouds could be a lack of biological INP at the time and location of our Arctic measurements as predicted in a model study by Wilson et al. (2015), so those clouds might not freeze at low temperatures (Shupe et al., 2008; Augustin-Bauditz et al., 2014).

The INP estimations for the Arctic (see Figure 13, left panel) have to be used with caution, because the 'out of cloud' probed altitude range only covers warm temperatures, where the INP estimation is not very sensitive to the measured aerosol concentrations. However, the generally indicated low INP concentrations might be reflect in the high fraction of 'Mostly liquid' clouds, which slightly decreases with decreasing temperature.

However, an inspection of the vertical velocities measured during the Arctic campaigns in Figure 12 (left panel) indicates that 60% of the 'mostly liquid' (pink) clouds are found in areas with very low updrafts, fluctuating around zero, while 40% are found in weak updrafts/downdrafts, respectively. Comparably weak updrafts are also frequently found in the 'WBF' (dark blue) clouds. This is to be expected, because the WBF regime develops in




weak updrafts, implying that the trigger to transform a cloud from liquid to ice is the available INP concentration. The 'Coexistence' (brown) and 'Secondary ice' clouds were observed with low frequency (<1%) in the Arctic and show a slightly wider spread in updraft velocities. In particular, higher updrafts occurred more often ($\sim$ 30%) in these clouds, which is consistent with the theoretical considerations shown in Figure 2 for the 'Coexistence'

regime.

### 3.5.2 Mid-latitude clouds

At mid-latitudes (COALESC field campaign), the largest cloud fraction are the fully glaciated WBF clouds (dark blue in Figure 15, middle panel). This is consistent with the assumption that at mid-latitudes, the WBF process is the dominant process for cloud evolution (Boucher et al., 2013). More INP seem to be available that are ice active

at and below -10 °C (263 K). At temperatures warmer than -20 °C (253 K), the fraction of this cloud type is slowly reduced, while more and more 'mostly liquid' clouds (pink in Figure 15) and coexistence clouds (brown in Figure 15) are found for higher temperatures. The WBF process depends on the presence of INP, which are observed in higher quantities at mid-latitudes in comparison to the Arctic (compare Figure 13). The varying occurrence of different cloud types with temperature - i.e. 'mostly liquid' clouds at higher temperatures (lower altitudes) and an

increasing part of 'WBF' clouds with decreasing temperature (increasing altitude) - might correspond to different INP regimes. At temperatures below about -20 °C, for example, efficient mineral dust INP initiate the freezing process, while at warmer temperatures less frequently occurring biological particles act as INP (Augustin-Bauditz et al., 2014; Kanji et al., 2017). In addition, the increasing fraction of 'WBF' clouds with decreasing temperature reflects the fact that the colder the environment is, the higher the probability is that the RHw falls below 100%:

with decreasing temperature, more and more droplets freeze and exploit the gas phase water when they grow. As a consequence, less gas phase water is available the colder the temperature is. In the transition range between predominantly 'mostly liquid' and only 'WBF' clouds (temperatures between -20 to -10 °C - 253 and 263 K:), 'Coexistence' clouds appear, which we interpret as clouds where the freezing process has started, but in which the RHw is still above 100% (blue curve slightly below RHw = 100% in Figure 2).

'Secondary ice' clouds appear in mid-latitude clouds more often than in the Arctic. It is unlikely that these small particles are shattering artifacts, because they often occur in clouds with no or few large ice particles - these large particles, however, are those that usually shatter (Korolev et al., 2011). In addition, as discussed in Section 2.3, based on IAT analysis shattering could be almost excluded in the measurements. In contrast, the majority of those clouds occur at temperatures between -5 to -13 °C (268 to 258 K), which is an indication for an efficient Hallett-

Mossop process having altered the cloud at slightly warmer temperatures. Note that the classification aims at the result of cloud transforming processes, not the cloud transformation itself. Which process precisely took place before the cloud section was probed can not be proven with this 1 Hz data set.

At mid-latitudes, 'mostly liquid', 'Coexistence' and 'WBF' clouds show the same vertical velocity distributions (Figure 12). The peak updrafts are slightly higher and the widths slightly narrower in comparison to the Arctic




clouds. This is another hint that underscores the above discussed dependence of the cloud categories on RHw: within the same vertical velocity range, the relative humidity can vary strongly depending on the available amount of water and the cloud development stage (cloud particle nucleation, sedimentation, evaporation). The 'Secondary ice' clouds show a different updraft distribution with faster vertical velocities, which might indicate that these
clouds occurred in more turbulent environments.

### 3.5.3   Tropical clouds

During the tropical field campaign ACRIDICON-CHUVA in convective towers, stronger updrafts and downdrafts were observed more frequently than during the other campaigns (Figure 12, right panel). The records show extreme vertical velocities up to $-10\,\mathrm{m\,s^{-1}}$ and $+15\,\mathrm{m\,s^{-1}}$. However, these events were rarely observed, because due to
flight safety, these cloud sections were mostly avoided. Velocities of $0.5\,\mathrm{m\,s^{-1}}$ to $1.0\,\mathrm{m\,s^{-1}}$ were observed in more than 10% of all data points. The wider distribution of vertical velocities shows that the cloud dynamics are much stronger in the tropical clouds than at mid-latitudes and in the Arctic.

In comparison to the other regions, less 'mostly liquid' clouds are found in the tropics, also for warmer temperatures. This indicates a higher concentration of INP that are already ice active at comparably high temperatures,
pointing at biological INP. This seems to be plausible for tropical regions, but is only partially confirmed by the INP estimate (see Figure 13, right panel), where it should be added that the parameterization of DeMott et al. (2010) rather aims at INP from mineral dusts. The probed clouds occurred in both very clean air with less INP than at mid-latitudes case and in heavily polluted areas over fire clearance regions. A more detailed study on how the aerosol concentration affects the cloud type distribution during ACRIDICON-CHUVA was done by Cecchini
et al. (2017), also based on NIXE-CAPS aspherical fractions. The study shows that clouds in polluted environments contained more and smaller liquid water droplets and less ice, while clouds in clean conditions held more ice crystals and few liquid water droplets.

As a consequence of the higher vertical velocities in the convective towers, more 'Coexistence' clouds are observed than at mid-latitudes or in the Arctic. A small part of the liquid droplets $< 50\,\mu m$ survived down to the
homogeneous drop freezing temperature ($\sim -38\,°C$) in cases where the vertical velocity was high enough (see also Figure 2, red).

However, the 'WBF/large ice' (Figure 15, right panel) clouds are the most frequent at all temperatures. Those large cloud particles might stem from sedimentation out of the cloud anvils, which usually consist of mostly large aggregates, or might be transported downwards in the strong downdrafts within the convective clouds (compare
Jäkel et al., 2017).

It is, nevertheless, important to note again that due to security restrictions, the in-situ measurements were mostly restricted to cloud regions with small updraft velocities (see Figure 12), i.e. to young developing clouds or edges of convective towers. Due to this flight pattern, we most probably have probed conditions that favor the WBF process (consistent with Figure 2, blue) even if those conditions might not be representative for tropical convective clouds



in general. This part of the analysis should therefore be seen as an incentive for further studies and not be used as a basis for cloud type statistics in tropical dry seasonal convection.

In the tropical dataset, the cloud type 'Secondary ice' is scarce at the lower levels - as at mid-latitudes - but prevalent at cold temperatures, i.e. at high levels. The high concentrations of small aspherical particles might
indicate a population of frozen droplets that quickly develop complex shapes in supersaturation. Alternatively, other ice multiplication processes (e.g. ice splintering) take place more frequently at later cloud development stages. Again, as discussed in Section 3.5.2, shattering artifacts can be almost excluded as the reason for the high number of small aspherical particles: Large ice crystals appear at all temperatures up to $0\,°C$; the 'secondary ice' cloud type is, however, only observed at temperatures between $-38\,°C$ and $-20\,°C$. Additionally, an analysis of
inter-arrival times of the 'Secondary ice' cloud sections did not show shorter inter-arrival times than in other parts of the dataset.

## 4   Summary and conclusions

The study presented here gives an overview of typical cloud properties observed between $0\,°C$ and $-38\,°C$ ('mixed-phase temperature regime') and links the clouds at differing stages of glaciation to ice formation and evolution
mechanisms. It gives hints to the relevance of cloud processes at different geographical locations and altitudes.

To this end, the cloud spectrometer NIXE-CAPS was deployed in four airborne field campaigns to conduct measurements of cloud particle sizes, number concentrations and, as an additional parameter, the cloud particles' asphericity. Based on the observations, which consist of 38.6 hours within clouds, we developed algorithms based on the measurements of particle size distributions and aspherical fractions to identify four cloud types:

– 'Mostly liquid': dense clouds consisting of mostly small droplets. All particles in the size range from $20\,\mu m$ to $50\,\mu m$ are spherical. The few large cloud particles $> 50\,\mu m$ might occasionally include ice crystals.

       – 'Coexistence': dense clouds consisting of mostly small particles with a low percentage ($< 50\%$) of small aspherical ice particles, ice crystals $> 50\,\mu m$ are present. The coexistence of liquid droplets and ice crystals is most probably due to supersaturation over both water and ice caused by higher vertical velocities.

– 'Secondary Ice': dense clouds consisting of mostly small particles between $3\,\mu m$ and $50\,\mu m$ with a high percentage ($> 50\%$) of aspherical ice particles. The aspherical fractions found are comparable to those of cirrus clouds; we thus conclude that these clouds are completely glaciated. The large cloud particles $> 50\,\mu m$ are also frozen. The ice crystal numbers exceed the expected ice nuclei concentrations by several orders of magnitude, which suggests that the small crystals result from secondary ice production. Small ice crystal
production by shattering can be almost excluded from IAT analysis of the specific situations.

       – 'WBF/Large ice': thin clouds with low number concentrations, whose mass distribution is dominated by large cloud particles $> 50\,\mu m$; the aspherical fractions of the small particles are high and the large particles





are frozen: these clouds are fully glaciated. The reduced number of small particles in comparison to the 'mostly liquid' clouds can be explained by the WBF process. However, from the asphericity detection it is obvious that small ice crystals are present in WBF clouds with higher concentrations than large ice crystals. Alternatively, these clouds might consist of sedimenting aggregates. The cloud particle number concentrations agree reasonably well with the estimated ice nuclei concentrations.

We quantified the occurrence of these cloud types for Arctic, mid-latitude and tropical regions, respectively.

For the Arctic, we observed mpt clouds for temperatures higher than -20 °C. The largest part were 'Mostly liquid' clouds, with a small percentage of 'Coexistence' and 'WBF/Large ice' clouds. We hypothesize that this cloud type distribution is a result of low concentrations of ice active INP, particularly biological INP, during our field campaign in the Arctic. This hypothesis is in agreement with the low INP concentrations found for this region in a modelling study by Wilson et al. (2015).

At mid-latitudes, mpt clouds down to -40 °C were probed, mostly in frontal systems with moderate updrafts between 0 and $0.5\,\mathrm{m\,s^{-1}}$. Here, the glaciated 'WBF/Large ice' clouds dominate most of the temperature range, pointing to a sufficient availability of INP. Only at temperatures warmer than -20 °C an increasing fraction of 'Coexistence' clouds and also 'Secondary ice' clouds were found. The temperature range for the 'Secondary ice' clouds is consistent with the preconditions for the Hallett-Mossop process.

In the tropics, mostly moderate, but also very strong vertical velocities were recorded. Correspondingly, the glaciated 'WBF/Large ice' clouds dominate the measurements over all temperature ranges, but also 'Coexistence' clouds are observed down to -40 °C. The supercooled liquid droplets freeze homogeneously when transported to higher altitudes. 'Secondary ice' clouds are observed at colder temperatures (higher altitudes) than at mid-latitudes, indicating that other ice splintering processes than the Hallett-Mossop process might be active here.

Pruppacher et al. (1998) summarize several studies that tracked (a) the percentage of clouds containing no ice crystals or (b) the percentage of clouds containing ice crystals as a function of temperature. Their findings agree well with our observations at mid-latitudes. It is noteworthy, however, that in none of the studies presented therein, liquid cloud fractions as high as observed during VERDI and RACEPAC were reported.

In general, the analysis of small cloud particle aspherical fractions advises against the assumption that all cloud particles smaller than 50 μm are liquid. On the contrary, small particles were frequently found to be aspherical. The aspherical particle fractions are an important parameter for the identification of the four cloud types investigated here. Observations that contain this information (e.g. Mioche et al., 2017) can be used to extend the cloud statistics presented here. In case no small particle shapes are available, particle size distributions can be used to differ between the Type 1 cloud group (mostly liquid/coexistence/secondary ice clouds) and the Type 2 clouds (WBF clouds - large ice). A sufficiently large data base would e.g. allow the quantification of the efficiency of the WBF process with regard to temperature and location. Along these lines, this study might serve as a starting point for a growing cloud type database in the mpt regime.





*Acknowledgements.* This work was supported by the Max Planck society, the DFG (Deutsche Forschungsgemeinschaft, German Research Foundation) Priority Program SPP 1294, the German Aerospace Center (DLR), and the FAPESP (São Paolo Research Foundation). Heike Wex is currently funded by DFG within the Ice Nuclei research UnIT (INUIT, FOR 1525), WE 4722/1-2. We thank Martin Schnaiter and Emma Jaervinen for the fruitful discussions during the RICE03 campaign.



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



**Figures**

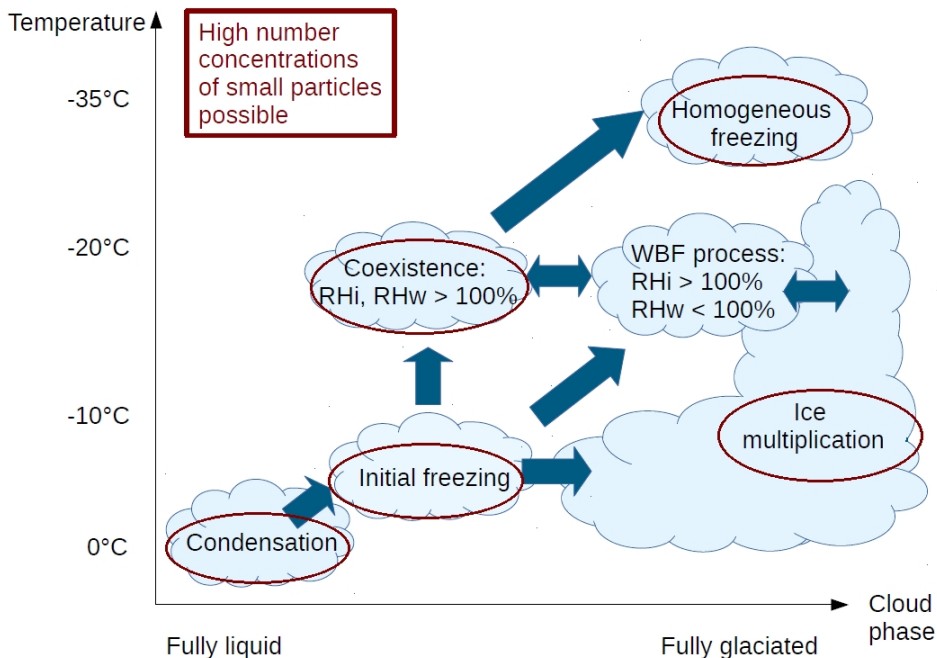

**Figure 1.** Possible paths to glaciation in the mixed-phase temperature regime





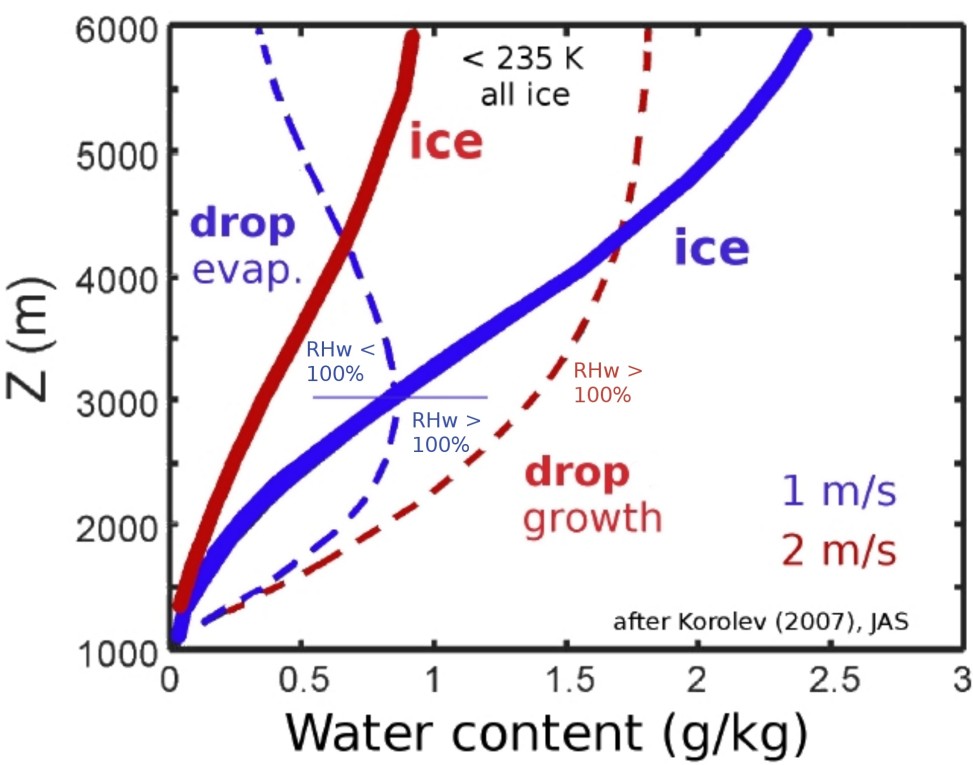

**Figure 2.** Liquid water content (dashed lines) and ice water content (solid lines) development with altitude (~ 1/temperature) in mixed-phase clouds for different vertical velocities (adapted from Korolev, 2007, with modification). Blue lines (updraft $1\,\mathrm{m\,s^{-1}}$): the cloud glaciates when RHw falls below 100% (WBF = Wegener-Bergeron-Findeisen regime); red lines (updraft $2\,\mathrm{m\,s^{-1}}$): RHw stays above 100%, liquid droplets and ice crystals coexist (Coexistence regime).





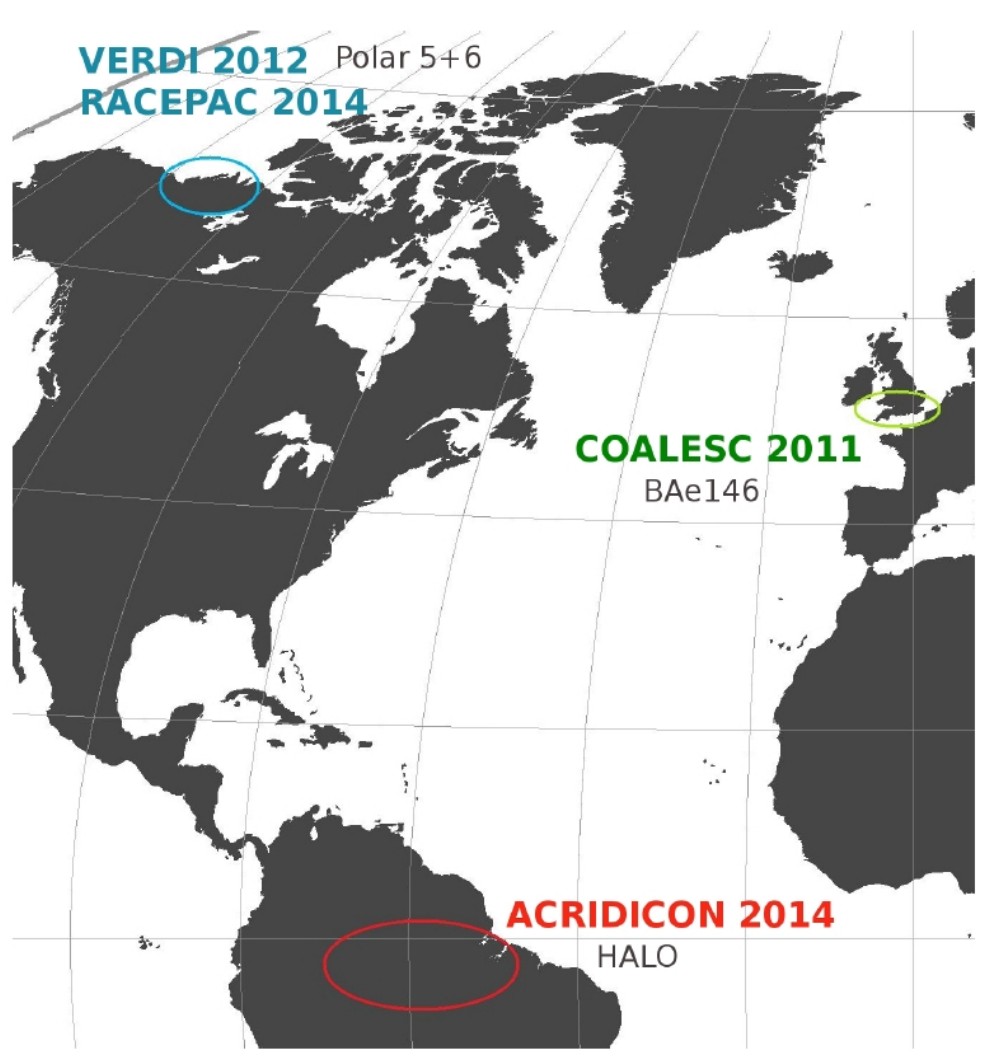

**Figure 3.** Locations of the campaigns comprised in this paper.

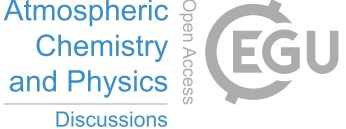

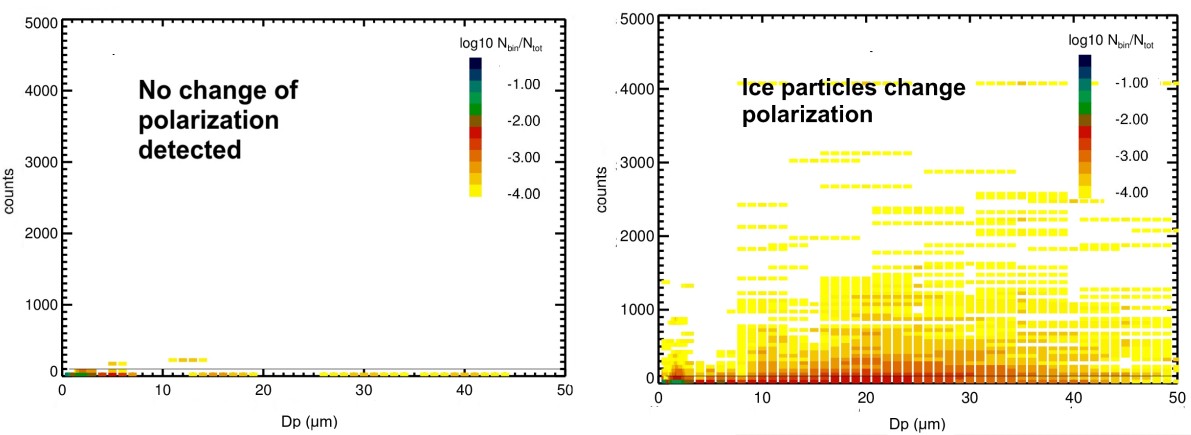

**Figure 4.** Left panel: Cross-polarized (S-pol) counts vs. particle size in a warm liquid cloud observed in the ACRIDICON-CHUVA campaign. The color code denotes the relative frequency of particles in this bin (Nbin) to overall particle count (Ntot). The spherical particles cause a weak signal in the S-pol detector. Right panel: Same, but in a cold cloud (-60 °C) consisting of ice crystals. Ice crystals can cause strong signals in the S-pol detector.





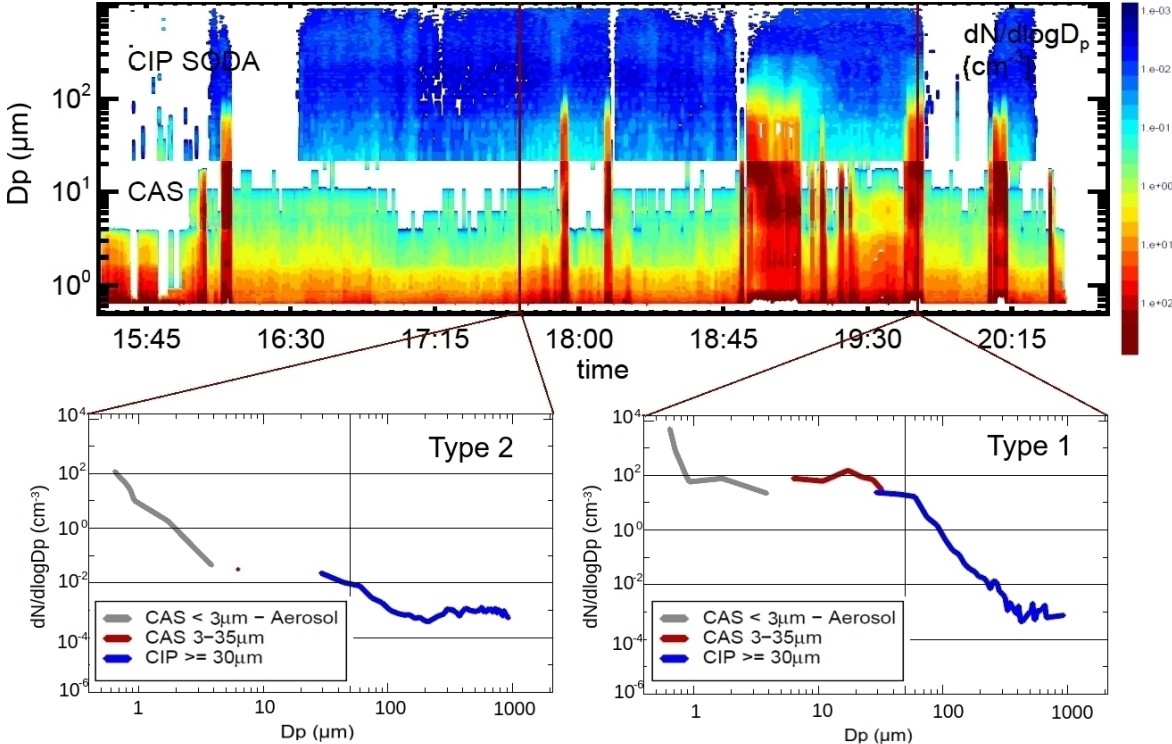

**Figure 5.** Size distributions along time during flight 08 of the VERDI campaign. Two types of clouds can be distinguished; one is dominated by the large particle mode (Type 2, example in lower left panel), the second by small particles (Type 1, example in lower right panel). The two cloud types are also associated with strongly differing particle number concentration ranges, cf. Figure 6.

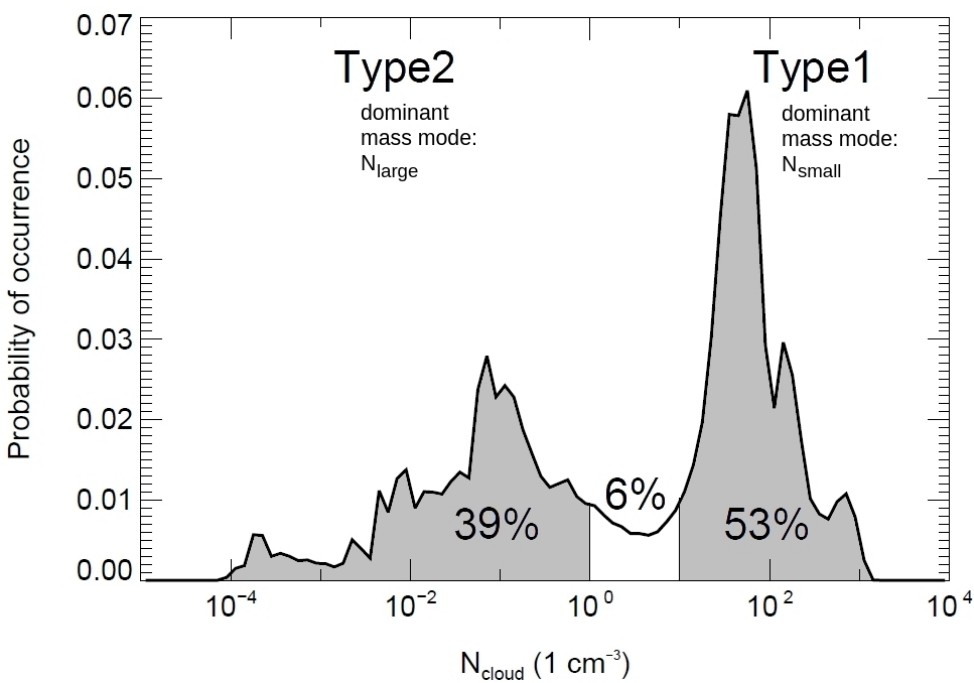

**Figure 6.** Histogram of cloud particle concentrations (Dp 3 $\mu$m to 937 $\mu$m) of Type 1 and Type 2 clouds in the mixed-phase temperature regime between 0 °C and -38 °C. For cloud type definitions see subsection 3.1. The 6% between the two clear modes were classified as 'Type 1' in this study. $N_{small}$: Particles with diameters between 3 $\mu$m and 50 $\mu$m. $N_{large}$: Particles with diameters > 50 $\mu$m. $N_{cloud}$: All particles with diameters of 3 $\mu$m and larger.





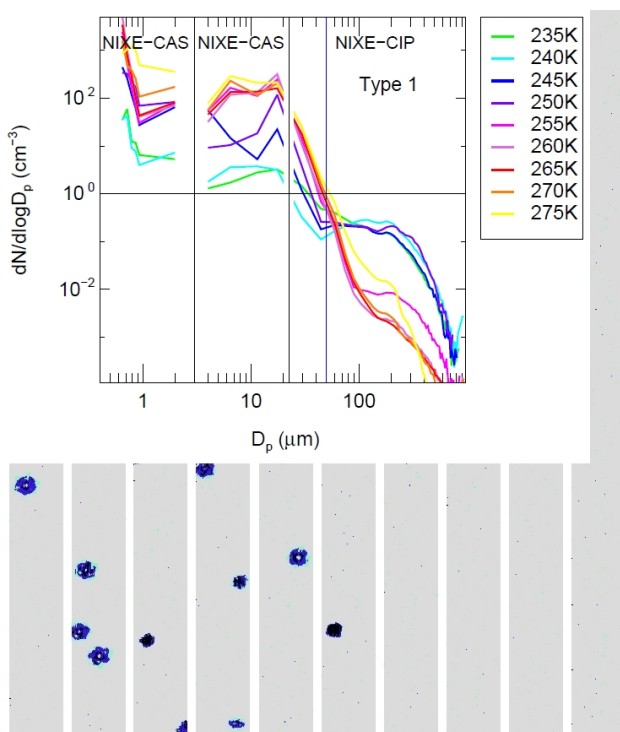

**Figure 7.** Type 1 clouds: Example of CIP images and average particle size distributions (PSDs) in 5 K intervals, all campaigns. The thin vertical line at 3 $\mu$m marks the boundary between aerosol and cloud particles. The line at 20 $\mu$m marks the transition from the NIXE-CAS-DPOL to the NIXE-CIPg instrument. The thick blue line divides the cloud particle population in particles smaller and larger than 50 $\mu$m.



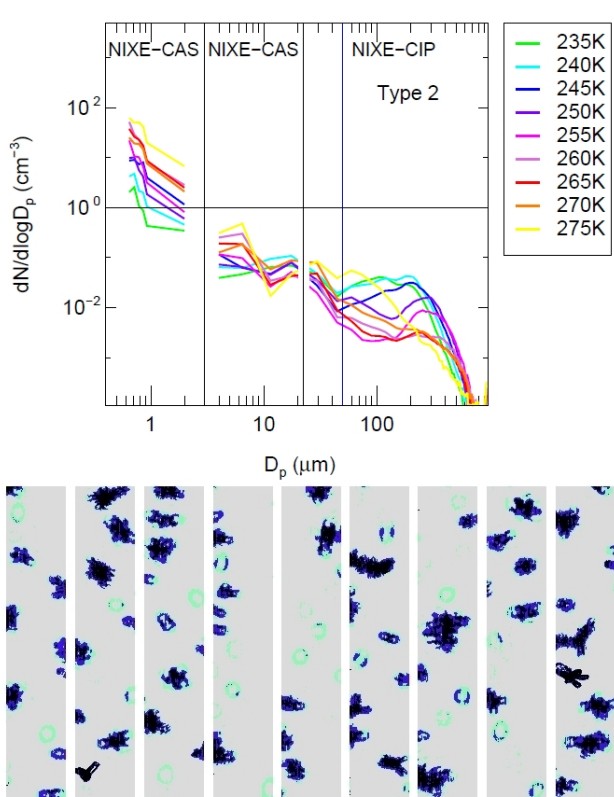

**Figure 8.** Same as in Figure 7, but for Type 2 clouds.




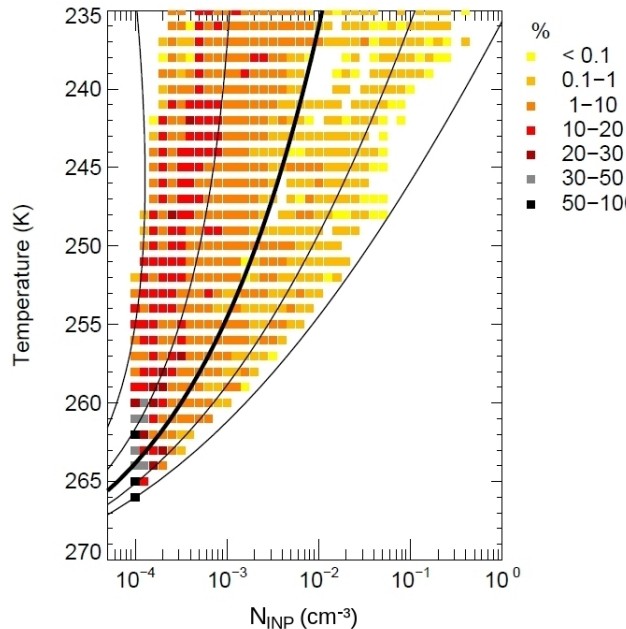

**Figure 9.** Frequencies of ice nucleating particle number concentrations ($N_{INP}$) vs. temperature for all measurement campaigns, estimated from NIXE-CAPS measurements of aerosol concentrations (Dp 0.6 - 3 $\mu$m) following DeMott et al. (2010). The black lines indicate INP concentrations for constant aerosol concentrations of 0.01 scm$^{-3}$ (leftmost line), 0.1 scm$^{-3}$, 1 scm$^{-3}$ (thick line), 10 cm$^{-3}$ and 100 scm$^{-3}$ (rightmost line).

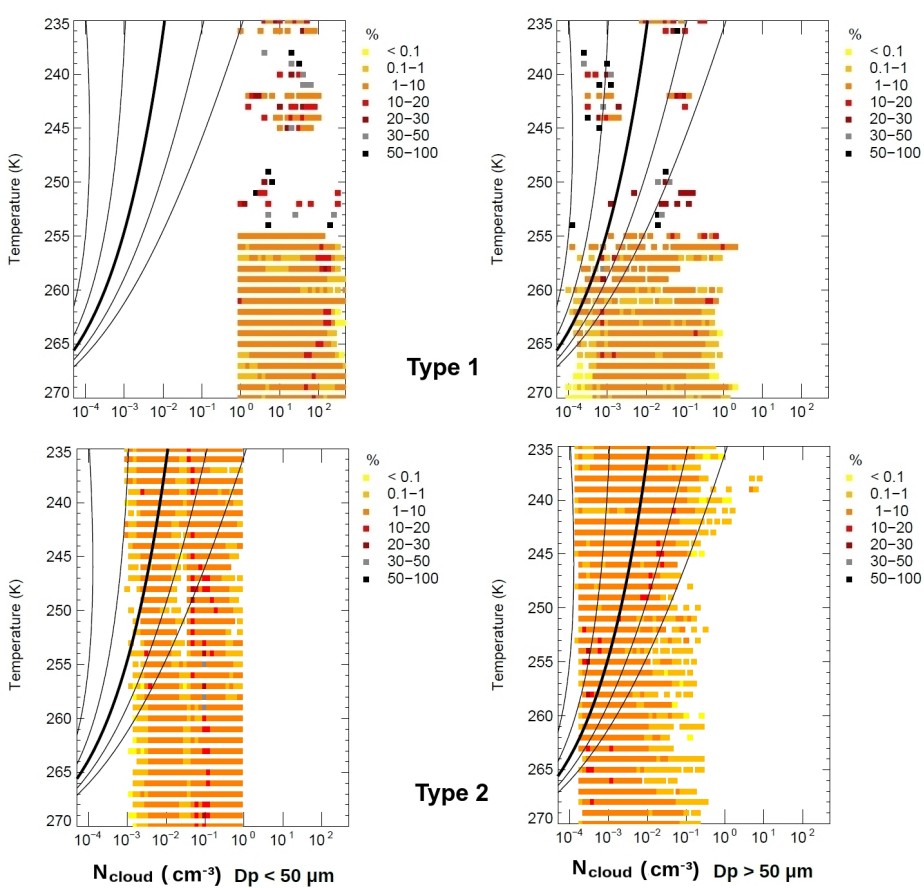

**Figure 10.** Same as Figure 9, but frequencies of cloud particle number concentrations for $N_{small}$ (left panel) and $N_{large}$ (right panel). Top row: Type 1 clouds, bottom row: Type 2 clouds.



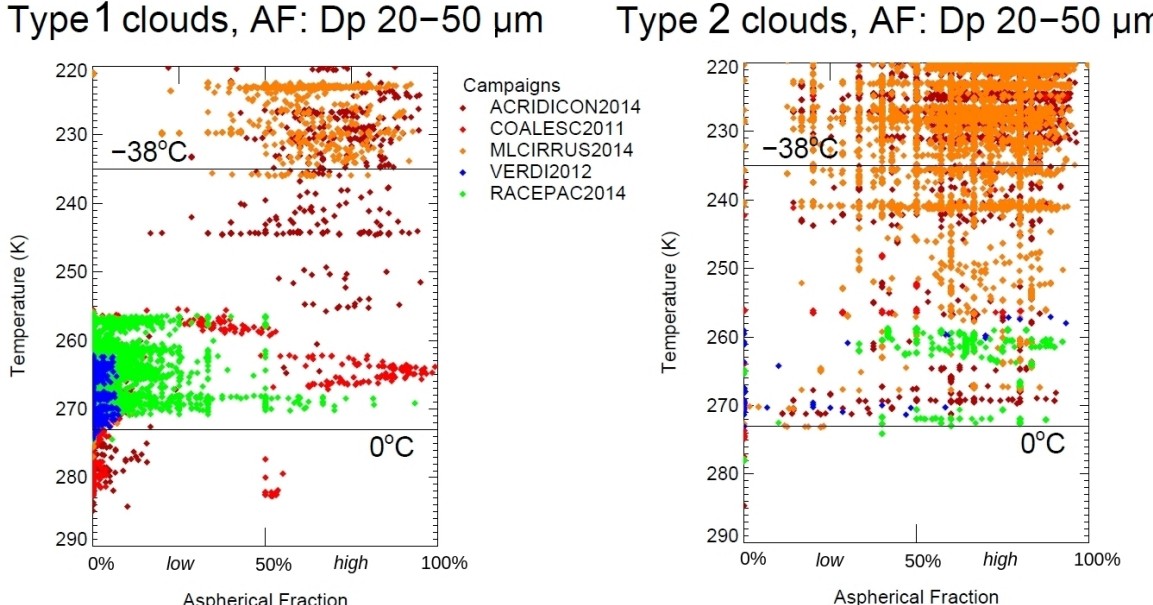

**Figure 11.** Aspherical fractions (AF) for Dp = 20 to 50 $\mu$m. Type 1 clouds show a variety of AF. Type 2 shows AFs comparable to cirrus clouds - which is illustrated by observations from the ML-Cirrus campaign - throughout the temperature range.



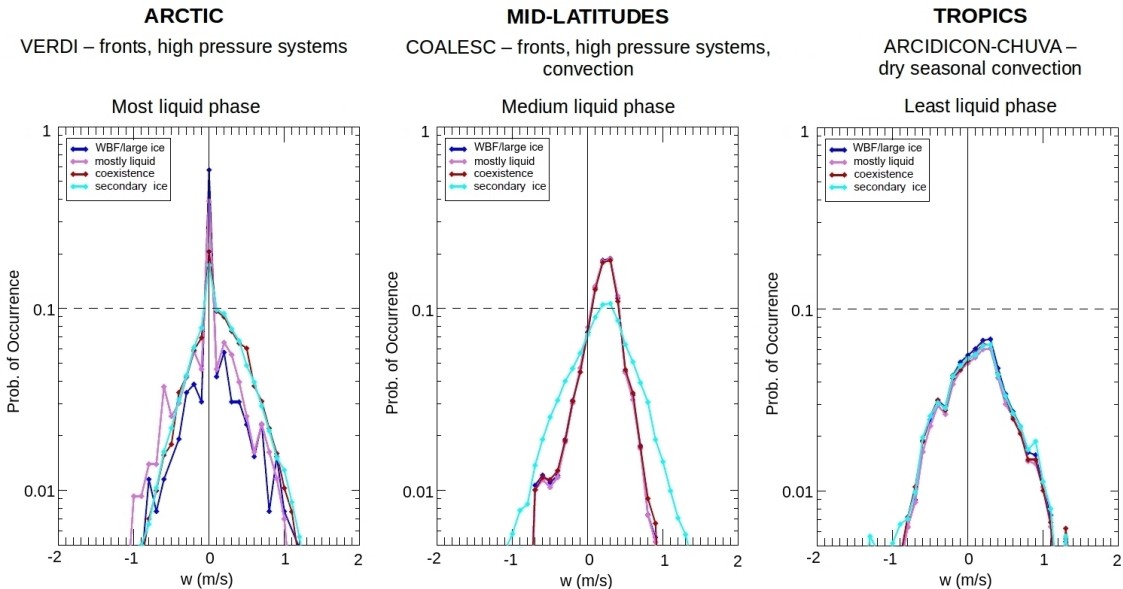

**Figure 12.** Frequency of occurrence for vertical velocities (w) within mpt clouds during the campaigns VERDI (Arctic), COALESC (mid-latitudes) and ACRIDICON-CHUVA (tropics).

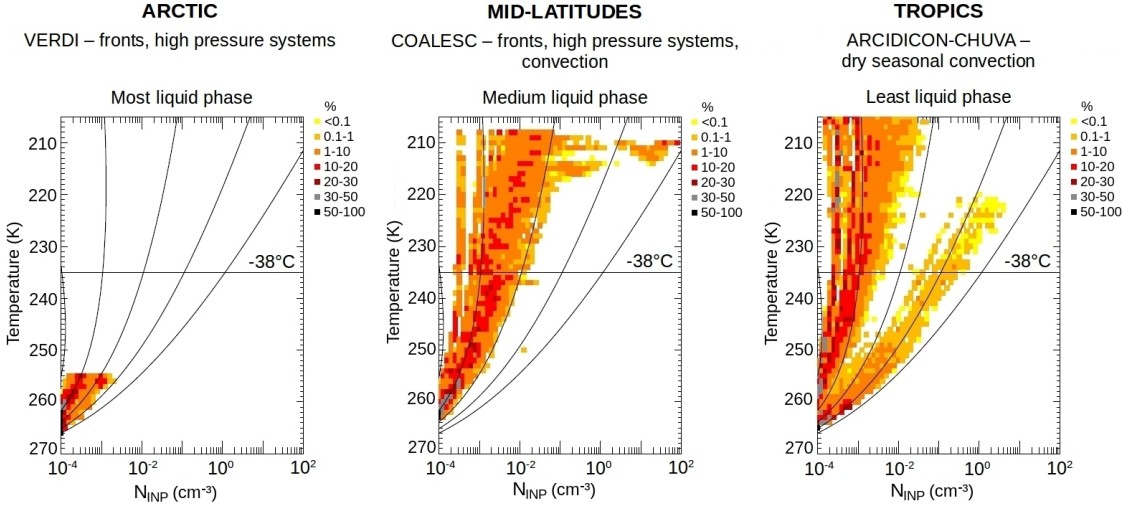

**Figure 13.** Frequencies of occurrence of INP concentrations ($N_{INP}$) vs. temperature during VERDI and RACEPAC (Arctic), COALESC (mid-latitudes) and ACRIDICON-CHUVA (tropics). INP number concentrations are estimated via aerosol concentrations for particles $> 0.6\,\mu$m following DeMott et al. (2010). The black lines indicate INP concentrations for constant aerosol concentrations of $0.01\,\mathrm{cm}^{-3}$ (leftmost line), $0.1\,\mathrm{cm}^{-3}$, $1\,\mathrm{cm}^{-3}$, $10\,\mathrm{cm}^{-3}$ and $100\,\mathrm{cm}^{-3}$ (rightmost line).




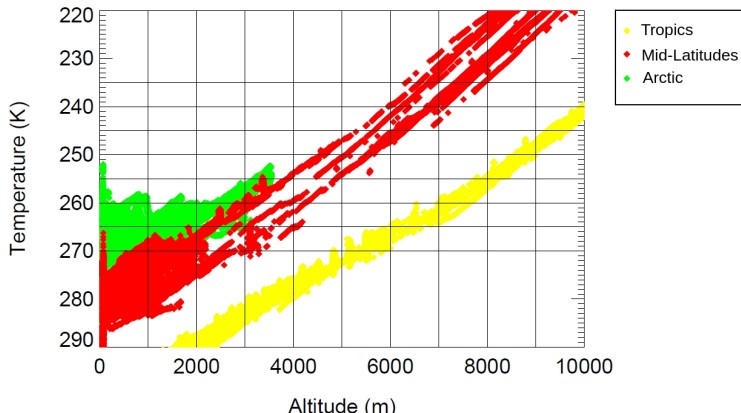

**Figure 14.** Temperature vs. altitude for the field campaigns VERDI and RACEPAC (Arctic), COALESC (Mid-Latitudes) and ACRIDICON-CHUVA (Tropics). The profile differs due to the varying latitudes.





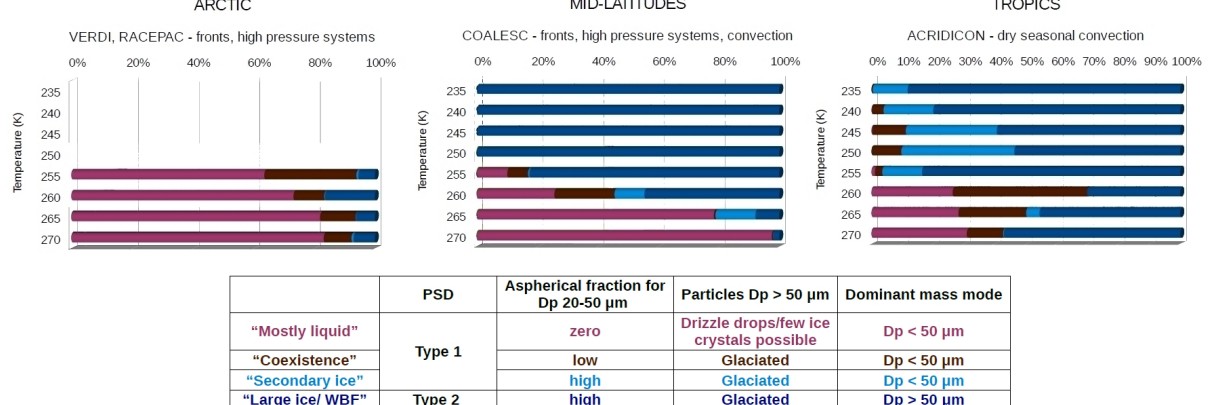

**Figure 15.** Occurrence of the cloud types defined in subsection 3.3: "Mostly liquid" clouds are dominated by small, exclusively spherical particles. They have high overall number concentrations. "Coexistence" clouds are dense, too, but do contain some small aspherical particles, indicating that a glaciation process has begun. The "secondary ice" cloud type is again very dense - the particle numbers exceed the INP concentration estimations by far (see subsection 3.2). Here, most of the small particles in the size range between $20\,\mu$m and $50\,\mu$m are aspherical; the cloud must therefore consist of ice. In contrast, clouds in the category "WBF/large ice" show low overall number concentrations. These clouds are dominated by large ice particles which may resume from the Wegener-Bergeron-Findeisen process or, especially in the tropics, be large, sedimenting ice aggregates from cumulonimbus anvils.





## Tables

| | Cloud particle concentration | Particles Dp < 50 µm | Particles Dp > 50 µm | Dominant mass mode |
|---|---|---|---|---|
| "Mostly liquid" | high | Liquid | Drizzle drops/few ice crystals possible | Dp < 50 µm |
| "Coexistence" | high | Mostly liquid, some ice crystals | Glaciated | Dp < 50 µm |
| "Secondary ice" | high | Glaciated | Glaciated | Dp < 50 µm |
| "Large ice/ WBF" | low | Glaciated | Glaciated | Dp > 50 µm |

**Table 1.** Characteristics of the cloud types expected in the mpt regime.

**Table 2.** Flight table for COALESC

| Date | Probed clouds - flight objectives | Cloud T in the mpt regime | Minutes in mpt clouds |
|---|---|---|---|
| 15.02.2011 | Warm clouds, mixed clouds, cirrus; test flight | -1.5 °C to -37.6 °C | 85.1 |
| 23.02.2011 | Warm clouds, cirrus clouds | 0 °C to -37.8 °C | 11.7 |
| 24.02.2011 | Warm Stratocumulus | 0 °C to -0.1 °C | 0.1 |
| 26.02.2011 | Stratocumulus in mixed-phase T regime | 0 °C to -17.9 °C | 46.0 |
| 01.03.2011 | Stratocumulus in mixed-phase T regime | 0 °C to -6.4 °C | 124.7 |
| 02.03.2011 | Stratocumulus | 0 °C to -3.1 °C | 92.0 |
| 03.03.2011 | Stratocumulus | 0 °C to -4.4 °C | 61.9 |
| 05.03.2011 | Stratocumulus | 0 °C to -3.3 °C | 51.4 |
| 07.03.2011 | No clouds | – | 0 |
| 08.03.2011 | Warm stratocumulus and cirrus clouds | 0 °C to -38.0 °C | 47.0 |
| 11.03.2011 | Stratocumulus | 0 °C to -4.9 °C | 105.9 |
| 14.03.2011 | Mostly cirrus clouds | -8.9 °C to -37.9 °C | 10.6 |
| 15.03.2011 | Stratocumulus and cirrus | 0 °C to -38.0 °C | 25.8 |
| 16.03.2011 | Stratocumulus | 0 °C to -0.3 °C | 6.7 |
| 18.03.2011 | No clouds | – | 0 |
| 19.03.2011 | Mostly contrail cirrus | -18.1 °C to -38.0 °C | 11.9 |





**Table 3.** Flight table for VERDI

| Date | Probed clouds - flight objectives | Cloud T in the mpt regime | Minutes in mpt clouds |
|------|-----------------------------------|---------------------------|-----------------------|
| 25.04.2012 | Low mostly liquid stratus, test flight | -3.7 °C to -9.1 °C | 47.1 |
| 27.04.2012 | Stratus (liquid and ice) over sea ice | -8.1 °C to -16.5 °C | 73.4 |
| 27.04.2012 | Low dissipating clouds over sea ice | -9.1 °C to -17.3 °C | 47.6 |
| 29.04.2012 | Stable stratus over sea ice | -8.4 °C to -12.5 °C | 77.9 |
| 30.04.2012 | Extensive cloud with layer structure | -6.3 °C to -19.1 °C | 212.8 |
| 03.05.2012 | Thin low subvisible clouds | -9.4 °C to -12.1 °C | 56.15 |
| 05.05.2012 | Patchy low cloud layer at mpt regime | -8.6 °C to -16.8 °C | 77.9 |
| 08.05.2012 | Mostly supercooled liquid clouds, two layers | -4.9 °C to -9.7 °C | 65.8 |
| 10.05.2012 | Dissolving altostratus layer | -5.5 °C to -11.2 °C | 45.1 |
| 14.05.2012 | Two thin stratus and cumulus | -1.4 °C to -5.8 °C | 41.9 |
| 15.05.2012 | Mostly liquid stratus and a cumulus | -0.7 °C to -14.1 °C | 73.2 |
| 16.05.2012 | Thin, mostly liquid stratus | -1.7 °C to -5.3 °C | 95.2 |
| 17.05.2012 | Mostly liquid stratus with large snow | 0 °C to -6.3 °C | 54.5 |



**Table 4.** Flight table for RACEPAC

| Date | Probed clouds - flight objectives | T range/cloud top T | Minutes in mpt clouds |
|------|-----------------------------------|---------------------|-----------------------|
| 28.04.2014 | Cumulus | -12.9 °C to -17.8 °C | 54.1 |
| 30.04.2014 | Low level clouds in cold sector of a low | -2.3 °C to -14.4 °C | 70.2 |
| 01.05.2014 | Thin fog layer | -2.0 °C to -9.6 °C | 5.0 |
| 03.05.2014 | Single/double layer liquid dominated cloud | 0 °C to -2.4 °C | 27.2 |
| 06.05.2014 | Single/multilayer clouds | 0 °C to -6.3 °C | 55.6 |
| 08.05.2014 | Thick stratus | 0 °C to -3.8 °C | 22.5 |
| 10.05.2014 | Two stratus clouds | -3.0 °C to -9.1 °C | 49.0 |
| 11.05.2014 | No clouds | – | 0 |
| 13.05.2014 | No clouds | – | 0 |
| 14.05.2014 | Homogeneous stratus | -1.9 °C to -10.1 °C | 25.8 |
| 16.05.2014 | Midlevel clouds | 0 °C to -10.1 °C | 75.7 |
| 17.05.2014 | Liquid and ice clouds on various altitudes | 0 °C to -11.3 °C | 22.7 |
| 20.05.2014 | Low-level clouds | -1.5 °C to -9.5 °C | 54.2 |
| 22.05.2014 | Low-level clouds before front | -6.1 °C to -15.0 °C | 29.2 |
| 22.05.2014 | Stratus behind front | -1.5 °C to -11.8 °C | 29.6 |
| 23.05.2014 | Midlevel clouds | -2.3 °C to -15.1 °C | 14.3 |





**Table 5.** Flight table for ACRIDICON-CHUVA

| Date | Probed clouds - flight objectives | Cloud T in the mpt regime | Minutes in mpt clouds |
|---|---|---|---|
| 06.09.2014 | Convective cloud profiling and outflow | 0 °C to -32.2 °C | 13.2 |
| 09.09.2014 | Convective cloud profiling | 0 °C to -1.2 °C | 1.1 |
| 11.09.2014 | Convective cloud profiling and outflow | 0 °C to -38.0 °C | 8.6 |
| 12.09.2014 | Cloud tops for satellite comparison | 0 °C to -29.6 °C | 5.5 |
| 16.09.2014 | Pyrocumulus profiling and outflow | 0 °C to -38.0 °C | 18.1 |
| 18.09.2014 | Shallow convective cloud profiling and outflow | -36.6 °C to -38.0 °C | 1.4 |
| 19.09.2014 | Pyrocumulus profiling, convective outflow | -0.4 °C to -35.1 °C | 8.8 |
| 21.09.2014 | Albedo flight | – | 0 |
| 23.09.2014 | Convective cloud profiling and outflow | 0 °C to -38.0 °C | 5.5 |
| 25.09.2014 | Cb anvil/outflow | -29.4 °C to -38.0 °C | 13.5 |
| 27.09.2014 | Warm clouds over forested and deforested areas | – | 0 |
| 28.09.2014 | Convective cloud profiling | 0 °C to -38.0 °C | 11.1 |
| 30.09.2014 | Albedo flight | – | 0 |
| 01.10.2014 | Convective cloud profiling and outflow | 0 °C to -5.6 °C | 2.6 |





| Type1 | $N_{small}$ (cm⁻³) | $N_{large}$ (cm⁻³) |
|---|---|---|
| 235 K | 2.207 | 0.162 |
| 240 K | 2.632 | 0.177 |
| 245 K | 19.894 | 0.134 |
| 250 K | 24.902 | 0.166 |
| 255 K | 109.944 | 0.035 |
| 260 K | 109.798 | 0.022 |
| 265 K | 269.979 | 0.032 |
| 270 K | 166.362 | 0.047 |
| 275 K | 67.788 | 0.098 |

| Type2 | $N_{small}$ (cm⁻³) | $N_{large}$ (cm⁻³) |
|---|---|---|
| 235 K | 0.057 | 0.023 |
| 240 K | 0.080 | 0.025 |
| 245 K | 0.069 | 0.017 |
| 250 K | 0.062 | 0.010 |
| 255 K | 0.064 | 0.004 |
| 260 K | 0.140 | 0.003 |
| 265 K | 0.070 | 0.003 |
| 270 K | 0.116 | 0.005 |
| 275 K | 0.117 | 0.017 |

**Table 6.** Average cloud particle concentrations for the two cloud types defined in subsection 3.1 (see also Figure 5), for both small ($D_p < 50\,\mu$m) and large ($D_p > 50\,\mu$m) cloud particles.