# Peer review of "Classification of Arctic, Mid-Latitude and Tropical Clouds in the Mixed-Phase Temperature Regime"

_Atmospheric Chemistry and Physics, 2017_

## Referee Comment (RC1) · Anonymous Referee #1 · 18 Apr 2017

This paper provides a statistical analysis of the microphysical properties of mixed phase clouds in three different areas of the globe: the tropics, the midlatitudes, and the Arctic. This paper is definitely a good fit for ACP and the results are timely, but I would like to see some concerns I have addressed before publication. The latter half of the introduction is worded in such a way that makes it look like there is no point to the study, which is clearly false. My largest concerns with the analysis are an inaccurate evaluation of the uncertainty due to shattered artifacts from the CIP probe and with a lack of discussion of the uncertainties associated with updraft velocity measurements, which can be on the order of 0.5 m/s for a given aircraft. The latter is especially important to the conclusions of the paper since the authors make several conclusions based

on p.d.f.s of vertical velocity that differ by less than this uncertainty.

Major concerns:

Page 4: The way the first paragraph is worded, a reader gets the impression that there are no advantages of using in situ observations over remote sensing for detecting the microphysical quantities of clouds since every technique out there has its flaws. It is not clear why you are using in situ measurements. For example, there are 4 sentences going into the flaws of shape identification algorithms in the first paragraph, and the CAS-DPOL is it's a good idea to say how the CAS-DPOL data helps to fill in the gap, but I think this should be emphasized more in this paragraph instead of 2 paragraphs down. Since the CAS-DPOL data in your paper are probably the most novel part of the paper, I would almost say that the limitations of current probes in identifying shape and the introduction of the CAS-DPOL and how it helps to provide a solution to this problem should be its own paragraph. I would also, after you mention the limitations of remote sensing measurements, go into some detail about how in situ measurements are the only direct way to measure the size, shape, and count of liquid and ice particles and are used to develop remote sensing retrievals. This would provide a better context as to why you specifically chose in situ measurements.

Lines 33-35, page 7: Why were the glaciated periods identified manually over using automated algorithms?

Line 12-14, page 8: While only 5% of the particles may be shattered artifacts as determined by IAT, these particles tend to be less than 500 microns in size. The sample volume of the CIP is significantly smaller for particles in this size range than for larger sizes, so any addition of smaller particles can easily change the number concentration by potentially a few hundred percent. Therefore, it is misleading to think that shattered artifacts would only have a 5% impact on the number concentrations. For example, Jackson et al. (2014) have shown that using IAT algorithms on 2DC probes with K-tips reduces the number concentration down by a factor of 2. Therefore, I think an analysis of how different the number concentrations are when the IAT algorithm is used and when it's not provides a better way to quantify the uncertainty due to shattered artifacts.

Line 5, page 9: This analysis really needs to go into more detail as to how the modes were determined, because while the authors identify two modes in their data, a reader can look at Figure 6 and see at least five, with three in the Type 1 region alone. How were the number of modes determined? How did you determine where the overlap region is between the modes? Does the smallest mode really have a peak at 10-4 cm-3? It looks like it's more around 10-1 cm-3 if the two modes are defined as they are in Figure 6.

Line 10, page 9: Figure 7 shows data from all campaigns, not just VERDI, since it does not look like there is any data at < 255 K for the Arctic in your dataset.

Line 29, page 10: I do not think that you can exclude primary nucleation as a source of the ice particles shown. What Figure 10 shows is that the ice particle concentrations exceed ice nuclei concentrations by orders of magnitude, which shows that secondary production is likely occurring. It does not, though, exclude primary nucleation from also contributing to the observed concentrations. However, it would be safe to assert that primary nucleation does not make a large contribution to the number of ice particles observed.

Lines 21-26, page 11: Do you have any more in depth statistics for the amount of irregular particles as a function of temperature for the Type 1 and Type 2 clouds. I think two extra panels on Figure 11 showing how many spherical vs. irregular particles you identified for each of the cloud types would be of great use.

Section 3.5.1. You go into x% are weak updrafts versus very low updrafts. To me, it looks like, in general, |w| < 1 m/s, indicating weak vertical motion throughout, which would be expected with stratiform clouds. Given that the uncertainty in measured w from aircraft is on the order of 0.5 m/s, I would argue that the difference between the four curves is within measurement uncertainty and that there are no real differences

between them. The same applies for the other two panels as well.

Page 13, lines 12-13: You can't determine this by looking at Figure 13 alone, since you have no observations < 255 K, and the observations of INP concentration at > 255 K actually look higher in the Arctic than in the midlatitudes.

Page 14, lines 1-5: I would argue all 4 vertical velocity p.d.f.s differ by less than the uncertainty in the measured vertical velocities, and hence, the differences seen are not statistically significant.

Figure 14: I honestly do not think this figure adds a whole lot to the paper outside of saying that the arctic is colder than the midlatitudes which is colder than the Tropics. I think this can generally be assumed and Figure 14 removed.

Minor changes:

Line 7 of Abstract: "clouds," should be "clouds;"

Lines 9-11: p.2. Run-on sentence. I would suggest fixing this up.

Line 15: p.2. "Formed" → "forms"

Figures 7/8: Scale needed for CIP images. Do the habits change with temperature? I think that information would be useful to provide. Also, 275 K is above freezing. Are you sure you are observing ice at that temperature?
* * *

---

## Referee Comment (RC2) · Anonymous Referee #2 · 12 Jun 2017

Review "Classification of Arctic, Mid-Latitude and Tropical Clouds in the Mixed-Phase Temperature Regime" by Anja Costa, Jessica Meyer, Armin Afchine, Anna Luebke, Gebhard Günther, James R. Dorsey, Martin W. Gallagher, André Ehrlich, Manfred Wendisch, Darrel Baumgardner, Heike Wex and Martina Krämer

In the manuscript "Classification of Arctic, Mid-Latitude and Tropical Clouds in the Mixed-Phase Temperature Regime" A. Costa et al. present a statistical classification of clouds in the mixed-phase temperature regime based on four extensive field campaigns in the Arctic, mid-latitudes and the tropics. The analysis is based on the measurement data from the cloud spectrometer NIXE-CAPS, which is the size distri-

bution in a wide size range and asphericity of the particles. The latter can be used to conclude the phase of the particles. The dataset is unique to quantify the microphysical characteristics of the investigated clouds in the mixed-phase temperature regime.

The clouds are divided in four cloud types in this study: liquid clouds, mixed-phase clouds, glaciated mixed-phase clouds due to the Wegener-Bergeron-Findeisen (WBF) process and clouds with secondary ice formation. In the Arctic the investigated clouds were mostly liquid, probably due to the shortage of ice nuclei. In the mid-latitudes the glaciated clouds due to the WBF process are dominant. At warm temperatures (in the temperature regime of the Hallet-Mossop process) some clouds with secondary ice formation occurred. In the tropics also glaciated mixed-phase clouds due to the Wegener-Bergeron-Findeisen (WBF) process were dominant throughout the whole temperature range. Secondary ice formation occur at much colder temperatures compared to the mid-latitudes.

The resulting cloud classification is very nice and provides some insights into the microphysics of mixed-phase clouds. The paper is written in a clearly structured way.

General comments:

- The uncertainty of the measurements, campaign etc. is not discussed very detailed. How sensitive might the results be to the time period and location of the campaign, e.g. weather conditions? How representative are the measurement campaigns of the general conditions in the areas? And assuming that the time period and location of the campaign itself is representative, how representative are the sampling times during a campaign, e.g. 1.5 h sampling during the ACRIDICON-CHUVA? It would be nice to add some more critical thoughts about this and maybe also a uncertainty estimation.

- Some microphysical processes occurring in mixed-phase clouds are strongly temperature dependent (freezing, secondary ice formation, but also accretion etc.). This could be included a bit more in the discussion, e.g. page 9, line 16-20.

[Figure]

- The mixture of naming the clouds Type 1 oder Type 2 clouds or referring to the micro-physics, e.g. 'Coexistence' clouds, 'WBF/Large ice' clouds is sometimes not so easy to follow. You could consider to always add a number to the cloud types, 1a, 1b, 1c and 2 (together with the physical naming).

- In the paper there are two different units used for the temperature, sometimes °C and sometimes K. At some parts that makes reading something out of the plot etc. quite difficult, e.g. page 10 line 24 this is difficult to read out of the plot right away, or Fig. 11 where you also have a mixture of °C and K. I would recommend to uniformly do everything using one unit.

- The term cloud particle is very general, sometimes it would be more specific to use the term cloud hydrometeors (to exclude aerosol particles).

Specific comments:

- page 2, line 11: Please also add a primary source.

- page 2, line 16-17: You are only referring to immersion freezing here. It should be legitimated or explained why or rephrased in a more general way.

- page 2, line 24: Be more clear what do you mean by high relative humidities (S=1 is sufficient for immersion freezing).

- page 3, line 13: Hallett-Mossop could be briefly explained (at least rime splintering could be mentioned).

- page 3, line 13 and line 28: Frozen droplet shattering is another process which can produce secondary ice (Mason and Maybank, 1960, Leisner et al. 2014, Lawson et al. 2015).

- page 3, line 14/15: Why only via contact freezing? The WBF process also takes place more efficient if there are more ice crystals in the close surroundings of supercooled water droplets.

- page 3, line 14/15: Contact freezing mostly refers to a collision of an aerosol particle with a supercooled droplet leading to freezing. In the case described the ice nucleus will be the ice crystal produced by secondary ice formation. An ice crystal is a perfect ice nucleus, but might be a bit confusing for some readers since ice nucleus is mostly associated with aerosol particles. Maybe it would be better to call it "collision freezing"?

- page 3, line 23: In case of sedimentation of ice particles it would not be a purely liquid cloud (or do the ice particles melt when falling into the cloud?).

- page 4, line 5: What is meant by active sensors? In-situ sensors?

- page 4, line 9: From which size on are they counted as ice particles?

- page 4, line 22: What is the lower threshold of the asphericity measurement of small particles?

- page 4, line 25: It is not clear here what is meant by "1 Hz data".

- page 4, line 32: How many clouds were sampled within these 38.6 hours? That would be a valuable and interesting information, especially in terms of the occurrence statistic in the end.

- page 5, line 32: Why was the flying speed of HALO so high? Is that related to the aircraft itself or due to meteorological conditions?

- page 5, line 11-12: Is the concentration limitation at high aircraft speed a problem? What are typical concentrations? How many particles are missed? That would be an interesting information to add.

- page 7, line 13: What is the consequence of the possibility of near spherical ice crystals? Is that accounted for in the uncertainty estimate and how?

- page 7, line 26: What is expressed by the shadow intensity?

- page 7, line 32: Why is the smaller particle fraction measured by the CAS? Is that

more sensitive in this size range compared to the CIP? How well do the number concentrations of both instruments overlap?

- page 8, line 8: Is the limitation to 300 particles per second reached often?

- page 8, line 11: What is an inter-arrival time correction? It would help the reader to understand the results if you add a one-sentence explanation here (it appears again in section 3.5.2).

- page 9, line 30: You could specify which lines the temperature groups refer to.

- page 9, line 32: This might be only true for the clouds where immersion freezing triggers the formation of ice crystals. That can be very different, especially for the convective tropical clouds, where ice crystals can sediment from colder regions of the cloud.

- page 10, line 4: The cloud particles are droplets or ice particles or both here?

- page 10, line 15: In DeMott et al. 2010 they do not limit the parameterization to 3 mum. The aerosol fraction estimated with NIXE-CAPS might therefore be underestimated (also because the lower threshold is at 0.6 mum instead of 0.5 mum). It would be nice if you could add some uncertainty estimation concerning this or some argumentation for your approach.

- page 10, line 15: The "aerosol data" used here are all particles in this size range so also all kind of hydrometeors? How good does that reflect the actual aerosol concentration? Are there some aerosol measurements for one of the four campaigns, where the approach could be evaluated against?

- page 10, line 19: How often is the maximum of $N\_INP$ reached and in which cases?

- page 10, line 33: "...formed around INP." might sound a bit miss leading, you could replace it by "...initial ice crystals have likely formed by immersion freezing" or "...and that the formation of the initial ice crystals has been likely initiated by INP immersed in

the cloud droplet".

- page 11, line 1: Could that problem be solved with a size-dependent ice nucleation parameterization? That would be a very interesting aspect to look at (if not in this paper then maybe discuss this shortly here).

- page 12, line 22: Were the campaigns mostly located over open water or ice? That could explain missing marine ice nuclei. However, the freezing efficiency of these ice nuclei is rather low compared to other aerosol species. Thus even if present it could be that the clouds might not freeze at low temperatures. You could use the parameterization given in Wilson et al. 2015 and the estimated aerosol concentration to check for a few cases how high the freezing probability would be.

- page 12, line 30: Can you give a range of values for "very low updrafts".

- page 13, line 12: The WBF depends only on the presence of INP in "classic" stratocumulus mixed-phase cloud cases. In convective tropical clouds it could also be triggered by sedimenting ice from colder parts of the cloud.

- page 13, line 13: It is actually difficult to see a strong difference in Fig. 13 if it is plotted like this. In the current representation it looks actually like the quantities are higher in the Arctic?

- page 13, line 17: Is that already clearly proven that biological particles occur less frequently compared to mineral dust? That might be different in the Arctic or over the Southern Ocean.

- page 13, line 25: Do you have a hypothesis why secondary ice clouds appear more often in the mid-latitudes compared to the Arctic?

- page 14, line 5: You could add another sentence as an explanation why secondary ice is more likely to form in turbulent environments.

- page 14, line 9-11: From Fig. 12 I can not recognize anything mentioned in the text

referring to the tropical vertical velocity, neither that they reach from -10 to -15 m sˆ-1, nor that velocities of 0.5 to 1 m sˆ-1 are reached in more than 10% of the cases, nor that the distribution if wider compared to the other cases. Maybe the plotting scale is wrong or the representation of the data inappropriate?

- page 14, line 15: It could point to biological INP, it could also point to strong sedimentation seeding the lower cloud levels.

- page 14, line 17: The focus of the DeMott et al. 2010 parameterization is not on dust. For the regions investigated all aerosol species are represented, which have an ice efficiency that leads to a frozen fraction larger than the detection limit of the instrument.

- page 16, line 3: Would it not be possible that these small ice crystals come from secondary ice formation as well?

- page 16, line 11: That (Wilson et al. 2015) might not be the best reference here- it would be better to cite ice nucleation field studies from the Arctic or the BACCHUS database, which was used in Wilson et al. 2015.

- page 16, line 22-25: It would be nice to have this aspect a bit more detailed, maybe adding the Pruppacher et al. estimates in Fig. 15 or have a separate figure for a comparison.

- Figure 1: The homogeneous freezing and ice multiplication cloud should be at the same location on the x-axis- both are fully glaciated.

- Figure 1: Coexistence is not really a path, it is clear from the x-axis that in this region there is coexistence. Maybe the different RHw areas could be colored in the background to also account for this aspect?

- Figure 1: Why is there an arrow pointing from Coexistence to homogeneous freezing clouds?

- Figure 1: Is there a reason for the WBF process to be located at -17°C?

- Figure 1: Especially in convective clouds instead of initial freezing there could be an interaction between the homogeneous freezing and the mpc cloud by sedimentation of ice crystals and thus a seeding of lower cloud regions.

- Figure 1: The different cloud types could be added in colors to the sketch.

- Figure 2: Where in the figure is (approx.) 235 K? Add a line to the corresponding altitude.

- Figure 2: Where exactly does the text < 235 K all ice refer to? Should not the drop growth curve then end at 235 K?

- Figure 4: What is the blue line in the plots?

- Figure 5: What does the color coding in the uppe panel stand for? dN/dlogDp?

- Figure 5: The labels of the color bar are too small.

- Figure 7 and 8: The second line is not at 20 mum.

- Figure 7 and 8: The lower panel is not explained in the caption or text. What is shown here? What does that show in addition to the size distributions? Do the stripes correspond to different diameters?

- Figure 9: The number of INP plotted nearly follows the constant aerosol concentration lines- does that only look like it or is the concentration not so variable with height/temperature?

- Figure 9: The color coding could be mentioned in the caption.

- Figure 9: Why is the temperature range limited here compared to Fig. 13?

- Figure 11: The Arctic campaigns look rather different compared to the other locations- why is that? Maybe also discuss that in the text with a reference to this figure.

- Figure 11: Why are there so few data points for the VERDI campaign?

- Figure 12: Why is the vertical velocity distribution from RACEPAC not added?

- Figure 13: Is the limitation of the data points in the Arctic case due to flight altitude?

- Figure 14: Is there not also a difference due to different flight altitudes?

- Table 1: Particles can not be glaciated, wording needs to be adapted.

- Table 2: What is the difference between a "Stratocumulus" and a "Stratocumulus in mixed-phase T regime"? Is there a certain temperature threshold (even below 0°C) assumed (row 5-7)? The cloud in row 1 is also likely to be "Stratocumulus in mixed-phase T regime"? Or why is it a "Warm cloud"?

- Table 4: Remove 11.05. and 13.05. or are the measurements done at these days used within this paper?

- Table 5: Remove 21.09., 27.09. and 30.09. or are the measurements used within this paper?

Small remarks,typos:

- page 1 line 3: Space missing between number and unit (to be consistent with the rest of the paper).

- page 1, line 7: Replace associated with by : .

- page 1, line 13: You might also want to specify the temperature range for the tropics.

- page 1, line 15: The second "to" is too much.

- page 2, line 21: Delete "nature of" (it is not the nature of the properties...).

- page 2, line 31: "with modification" is redundant (already written "adapted from").

- page 3, line 29: Verb missing.

[Figure]

- page 4, line 16/17: The clouds after the WBF process could eventually be named as 'glaciated clouds' (also this can be a bit ambiguous) or 'WBF glaciated clouds'. Or maybe it would be good to introduce the names here that are later on used, i.e. page 11 and 12.

- page 4, line 22: Add "(cloud spectrometer)".

- page 4, line 18-21: Sentence is too long and difficult to read.

- page 6 line 8: It would be nice to add a reference where NIXE-CAS-DPOL and NIXE-CIPg are explained later on in the text.

- page 6, line 13: One bracket too much.

- page 8, line 5: What does the "With these" refer to?

- page 9, line 10: One bracket too much.

- page 11, line 21: Shift this sentence to the beginning of this section.

- page 12, line 27: "reflected" instead of "reflect".

- page 12, line 24: The reference Augustin-Bauditz et al., 2014 does not fit in the context.

- page 13, line 10: Delete "and".

- page 13, line 22: replace the "-" with something equivalent, it could look like -253 K.

- page 13, line 22: There is a ":" too much.

- page 14, line 27: Switch bracket and "clouds".

- page 14, line 34: "darkblue" instead of "blue".

- page 15, line 27: "On the contrary" does not make sense since the statement is further supported?

- Caption Fig. 1: . missing at the end.

- Caption Fig. 2: Add "z" after altitude.

- Caption Fig. 2: Remove the bracket WBF..., that is not written in the figure.

- Figure 4: There are some black dots around the axis labels at the right panel of the figure.

- Figure 6: Delete the 1 in the unit-brackets.

- Figure 7 and 8: The blue line is not thick and not so easy to differentiate from the others.

- Figure 13: RACEPAC is mentioned in the caption but not in the title of the figure.

- Figure 15: The fonts are quite small.

- Table 5: To be consistent remove "profiling".

- Table 5: Write out "Cb".

---

## Author Comment (AC1) · 7 Aug 2017

**Classification of Arctic, Mid-Latitude and Tropical Clouds in the Mixed-Phase Temperature Regime**

Anja Costa[1], Jessica Meyer[1,2], Armin Afchine[1], Anna Luebke[1,3], Gebhard Günther[1], James R. Dorsey[4], Martin W. Gallagher[4], Andre Ehrlich[5], Manfred Wendisch[5], Darrel Baumgardner[6], Heike Wex[7], and Martina Krämer[1]

[1]Forschungszentrum Jülich GmbH, Jülich, Germany
[2]now at: Bundesanstalt für Arbeitsschutz und Arbeitsmedizin, Dortmund, Germany
[3]now at: Max Planck Institute for Meteorology, Atmosphere in the Earth System Department, Hamburg, Germany
[4]Centre for Atmospheric Science, University of Manchester, UK
[5]Leipziger Institut für Meteorologie, Universität Leipzig, Germany
[6]DMT, Boulder/Colorado, USA
[7]Leibniz Institute for Tropospheric Research, Leipzig, Germany
*Correspondence to:* Martina Krämer (m.kraemer@fz-juelich.de)

This document contains the comments of and answers to referee 1. Referee comments are marked in blue. Changes in the paper text are given in green.

Page 4: The way the first paragraph is worded, a reader gets the impression that there are no advantages of using in situ observations over remote sensing for detecting the microphysical quantities of clouds since every technique out there has its flaws. It is not clear why you are using in situ measurements. For example, there are 4 sentences going into the flaws of shape identification algorithms in the first paragraph, and the CAS-DPOL is it's a good idea to say how the CAS-DPOL data helps to fill in the gap, but I think this should be emphasized more in this paragraph instead of 2 paragraphs down. Since the CAS-DPOL data in your paper are probably the most novel part of the paper, I would almost say that the limitations of current probes in identifying shape and the introduction of the CAS-DPOL and how it helps to provide a solution to this problem should be its own paragraph. I would also, after you mention the limitations of remote sensing measurements, go into some detail about how in situ measurements are the only direct way to measure the size, shape, and count of liquid and ice particles and are used to develop remote sensing retrievals. This would provide a better context as to why you specifically chose in situ measurements.

Thank you for your comment, We have restructured the first section in the following way:

...Usually, they require a minimum number of pixels (corresponding to cloud particles with diameters of $70\,\mu$m and more) to recognize round or aspherical particles reliably. Due to these limitations, the shape identification of small particles has not been considered in many microphysical cloud studies. In the paper presented here, we use a new detector that can measure the asphericity of small ($< 50\,\mu$m) cloud particles (Baumgardner et al., 2014) together with a visual shape inspection of particles $> 50\,\mu$m. We thus hope to provide new insights into the

microphysical evolution of clouds in the mpt regime. To this end, we use in situ airborne cloud measurements in the cloud particle size range from 3 $\mu$m to 937 $\mu$m to classify the above described types of clouds in the mpt regime (see Figure 1): 'Mostly liquid' clouds after drop formation, 'coexistence clouds' after initial freezing, 'secondary ice' clouds influenced by ice multiplication, and clouds after the WBF process. This classification enables us to revisit a statistical overview published by Pruppacher et al. (1998), stating at which temperatures purely liquid or ice-containing clouds were found.....

Lines 33-35, page 7: Why were the glaciated periods identified manually over using automated algorithms?

The CIP probe records images in three shade intensities. The choice which shade intensity pixels are considered to be part of the particle image can influence the percentage of detected irregular particles by shape analysis algorithms. In particular, large ice crystals with several fully and several slightly shaded pixels can be erroneously identified as 'several small spherical particles'. Using all, i.e. also the slightly shaded pixels bears the risk of classifying out-of-focus droplets as irregular large ice crystals. The manual identification, on the other hand, allows to make these distinctions in a fast and easy way.

Line 12-14, page 8: While only 5% of the particles may be shattered artifacts as determined by IAT, these particles tend to be less than 500 microns in size. The sample volume of the CIP is significantly smaller for particles in this size range than for larger sizes, so any addition of smaller particles can easily change the number concentration by potentially a few hundred percent. Therefore, it is misleading to think that shattered artifacts would only have a 5% impact on the number concentrations. For example, Jackson et al. (2014) have shown that using IAT algorithms on 2DC probes with K-tips reduces the number concentration down by a factor of 2. Therefore, I think an analysis of how different the number concentrations are when the IAT algorithm is used and when it's not provides a better way to quantify the uncertainty due to shattered artifacts.

Our assumption is based on internal quality checks. To illustrate this, we attach a histogram which shows how strongly the IAT algorithm alters the CIP concentrations for ACRIDICON, the campaign with the largest particles and the highest aircraft speed (Figure 1). In most cases, no deviation for the CIP concentration with or without IAT algorithm is found. In addition, deviations stronger than a few percent are rare.

Line 5, page 9: This analysis really needs to go into more detail as to how the modes were determined, because while the authors identify two modes in their data, a reader can look at Figure 6 and see at least five, with three in the Type 1 region alone. How were the number of modes determined? How did you determine where the overlap region is between the modes? Does the smallest mode really have a peak at 10-4 cm-3? It looks like it's more around 10-1 cm-3 if the two modes are defined as they are in Figure 6.

[Figure]

**Figure 1.** Frequency of occurrence of deviations between IAT corrected and uncorrected CIP data for ACRIDICON-CHUVA 2014.

Thank you for pointing out this slightly unclear paragraph. We mention two modes in the beginning and then discuss three. This will be changed as shown below.

With regard to the determination of the two main modes, they are found via smoothing the histogram by taking into account the measurement uncertainties as discussed in section 2.2. With a concentration uncertainty of 20%,

5   only the two main modes remain which cover 39% and 53% of the dataset, respectively, and the slightly elevated frequency of occurrence of very low concentrations, which we trace back to the CIP detection limit.

In Figure 6, ... In this study, these measurements were assigned to Type 1 clouds. In addition to the two modes, a small peak at very low cloud particle concentrations (about $10^{-4}\,\mathrm{cm}^{-3}$) indicates slightly elevated concentrations around the detection limit of the CIP (a total of 5% of all observations). ....

10   Line 10, page 9: Figure 7 shows data from all campaigns, not just VERDI, since it does not look like there is any data at < 255 K for the Arctic in your dataset.

Thank you, this is corrected now.

Type 1 cloud characteristics measured during all campaigns described in section 2.1 are shown in Figure 7.

Line 29, page 10: I do not think that you can exclude primary nucleation as a source of the ice particles shown. What Figure 10 shows is that the ice particle concentrations exceed ice nuclei concentrations by orders of magnitude, which shows that secondary production is likely occurring. It does not, though, exclude primary nucleation from also contributing to the observed concentrations. However, it would be safe to assert that primary nucleation does not make a large contribution to the number of ice particles observed.

Thank you, we will follow your suggestion and change the text to:

In general, we can exclude primary ice nucleation as a main contributor for cloud particles in the Type 1 clouds.

Lines 21-26, page 11: Do you have any more in depth statistics for the amount of irregular particles as a function of temperature for the Type 1 and Type 2 clouds. I think two extra panels on Figure 11 showing how many spherical vs. irregular particles you identified for each of the cloud types would be of great use.

With respect to the small particle fraction, the shown data are all we can provide. Figure 11 shows the aspherical fractions, e.g. the percentage of aspherical to all observed small particles for all campaigns. The cloud types are defined via these aspherical fractions, i.e. you can read in the figure what the identified aspherical fractions e.g. for 'secondary ice' clouds at 265 Kelvin were. Figure 15 provides additional information on how often this cloud type occurred at the respective temperature.

Section 3.5.1. You go into x% are weak updrafts versus very low updrafts. To me, it looks like, in general, |w| < 1 m/s, indicating weak vertical motion throughout, which would be expected with stratiform clouds. Given that the uncertainty in measured w from aircraft is on the order of 0.5 m/s, I would argue that the difference between the four curves is within measurement uncertainty and that there are no real differences between them. The same applies for the other two panels as well.

Thank you for this remark. The uncertainties with regard to the vertical velocity measurements need to be pointed out. We have therefore added a comment (see below). In general, we think that the vertical velocity measurements should be shown despite the non-negligible uncertainties. While single data points might contain large errors, the fact that the distribution is smooth and centred near zero indicates that larger systematic differences between cloud types or campaigns should be visible, if they exist. Another interesting point to see is the rare occurrence of large vertical velocities during the tropical campaign, which holds true even for uncertainties of 0.5 m/s.

...which is consistent with the theoretical considerations shown in Figure 2 for the 'Coexistence' regime. Note that due to large uncertainties in the vertical velocity measurements, the statistical differences found between the cloud types should be regarded as an incentive for future investigations. While single data points might thus contain measurement errors, the distribution of observed vertical velocities is smooth and centred near zero, which

is expected for the meteorological situations discussed in section 2.1. Due to this and because our dataset consists of a large number of observations, we would like to point out the systematic differences found between cloud types and campaigns.

Page 13, lines 12-13: You can't determine this by looking at Figure 13 alone, since you have no observations < 255 K, and the observations of INP concentration at > 255 K actually look higher in the Arctic than in the midlatitudes.

Thank you for pointing this out. We've changed the text to:

The WBF process depends on the presence of INP, which are likely available in higher quantities at mid-latitudes in comparison to the Arctic (compare section 3.5.2 and Figure 13).

Page 14, lines 1-5: I would argue all 4 vertical velocity p.d.f.s differ by less than the uncertainty in the measured vertical velocities, and hence, the differences seen are not statistically significant.

Thank you for this remark, we've emphasized the uncertainties in section 3.5.1 (see above).

Figure 14: I honestly do not think this figure adds a whole lot to the paper outside of saying that the arctic is colder than the midlatitudes which is colder than the Tropics. I think this can generally be assumed and Figure 14 removed.

We agree that this is basic knowledge. However, since these temperature distributions have a strong effect on the actual spatial extent of the temperature range discussed throughout this paper, we feel that we need to make sure that readers keep these differences in mind. On former presentations, this Figure was explicitly requested by the audience. We would therefore like to keep it.

Minor changes: Lines 9-11: p.2. Run-on sentence. I would suggest fixing this up. Line 15: p.2. "Formed" ! "forms"

Thank you, the respective parts were corrected.

Text changes: see paper

Figures 7/8: Scale needed for CIP images. Do the habits change with temperature? I think that information would be useful to provide. Also, 275 K is above freezing. Are you sure you are observing ice at that temperature?

A scale was added. We did not perform a habit analysis for the presented study; this kind of analysis has not been done yet for the NIXE-CAPS dataset. This might be considered for a future study on the investigated cloud types.

With regard to your second concern, yes, there is clearly ice, often even large crystals. We assume that precipitating ice does not melt instantly when falling into regions slightly warmer than 273 K. When probing low cloud edges of precipitating clouds, it is therefore likely to find ice at these temperatures.

**References**

Baumgardner, D., Newton, R., Krämer, M., Meyer, J., Beyer, A., Wendisch, M., and Vochezer, P.: The Cloud Particle Spectrometer with Polarization Detection (CPSPD): A next generation open-path cloud probe for distinguishing liquid cloud droplets from ice crystals, Atmospheric Research, 142, 2 – 14, doi:http://dx.doi.org/10.1016/j.atmosres.2013.12.010, http://www.sciencedirect.com/science/article/pii/S0169809513003591, the 16th International Conference on Clouds and Precipitation, 2014.

Pruppacher, H. R., Klett, J. D., and Wang, P. K.: Microphysics of clouds and precipitation, Taylor & Francis, 1998.

---

## Author Comment (AC2) · 7 Aug 2017

**Classification of Arctic, Mid-Latitude and Tropical Clouds in the Mixed-Phase Temperature Regime**

Anja Costa[1], Jessica Meyer[1,2], Armin Afchine[1], Anna Luebke[1,3], Gebhard Günther[1], James R. Dorsey[4], Martin W. Gallagher[4], Andre Ehrlich[5], Manfred Wendisch[5], Darrel Baumgardner[6], Heike Wex[7], and Martina Krämer[1]

[1]Forschungszentrum Jülich GmbH, Jülich, Germany
[2]now at: Bundesanstalt für Arbeitsschutz und Arbeitsmedizin, Dortmund, Germany
[3]now at: Max Planck Institute for Meteorology, Atmosphere in the Earth System Department, Hamburg, Germany
[4]Centre for Atmospheric Science, University of Manchester, UK
[5]Leipziger Institut für Meteorologie, Universität Leipzig, Germany
[6]DMT, Boulder/Colorado, USA
[7]Leibniz Institute for Tropospheric Research, Leipzig, Germany

*Correspondence to:* Martina Krämer (m.kraemer@fz-juelich.de)

This document contains the comments of and answers to referee 2. Referee comments are marked in blue. Changes in the paper text are given in green.

The uncertainty of the measurements, campaign etc. is not discussed very detailed. How sensitive might the results be to the time period and location of the campaign, e.g. weather conditions? How representative are the measurement campaigns of the general conditions in the areas? And assuming that the time period and location of the campaign itself is representative, how representative are the sampling times during a campaign, e.g. 1.5 h sampling during the ACRIDICON-CHUVA? It would be nice to add some more critical thoughts about this and maybe also a uncertainty estimation.

The spatial and temporal coverage of the campaigns is of course small. We agree that the observations made in this study can only be evaluated in the context of the respective conditions. As stated in the paper, our conclusions relate to the conditions of Arctic and mid-latitude spring conditions and dry seasonal convection above the Amazon basin. The paper demonstrates how the new, easy-to-apply sorting algorithms can be used to reveal large differences between the campaigns with regard to the cloud processes. Readers should not be tempted to generalize our findings to all clouds in the respective geographical zone. As emphasized in abstract and summary, we hope to start a large database, which can then be used to answer this important question.

Some microphysical processes occurring in mixed-phase clouds are strongly temperature dependent (freezing, secondary ice formation, but also accretion etc.). This could be included a bit more in the discussion, e.g. page 9, line 16-20.

We agree that the temperature dependence needs to be discussed when assessing the occurrence of the different cloud types. We have chosen a later section, 3.5, for this discussion, because in this section, the above mentioned types are defined and identified in our dataset, so that they can be correlated to observed temperatures.

The mixture of naming the clouds Type 1 or Type 2 clouds or referring to the microphysics, e.g. 'Coexistence' clouds, 'WBF/Large ice' clouds is sometimes not so easy to follow. You could consider to always add a number to the cloud types, 1a, 1b, 1c and 2 (together with the physical naming).

Thank you, we followed your recommendation.

Text changes are all over the paper

In the paper there are two different units used for the temperature, sometimes degC and sometimes K. At some parts that makes reading something out of the plot etc. quite difficult, e.g. page 10 line 24 this is difficult to read out of the plot right away, or Fig. 11 where you also have a mixture of degC and K. I would recommend to uniformly do everything using one unit.

Thank you for this remark. Many publications describe the respective cloud processes in degC, while K is the SI unit. To account for both practices, we have now added both numbers to the text.

The term cloud particle is very general, sometimes it would be more specific to use the term cloud hydrometeors (to exclude aerosol particles).

Thank you for this recommendation. We now clarified in the methodology section (2.2, NIXE-CAPS description) that we use the terms 'hydrometeors' and 'cloud particles' synonymously for all particles larger than 3 micrometers.

page 2, line 11: Please also add a primary source

Ehrlich et al. (2009) present in-situ and remote sensing data from a campaign in the Arctic that show differences with respect to the absorbed solar radiation depending on the particle phase. Additionally, we have now included Curry et al. (1996) and Shupe and Intrieri (2004).

page 2, line 16-17: You are only referring to immersion freezing here. It should be legitimated or explained why or rephrased in a more general way.

Thank you for this remark, we have rephrased the sentence to:

initial freezing can occur, e.g. in those droplets that contain or touch an ice nucleating particle

page 2, line 24: Be more clear what do you mean by high relative humidities (S=1 is sufficient for immersion freezing).

We've added a reference to Kanji et al. for a more precise discussion of the favorable conditions. If we would specify the relevant humidity range, we would also need to define the respective temperature ranges, freezing mechanisms, INP types and so on, which we feel is not a simplified summary and would exceed the range of our paper.

The nature of the INP properties that favor ice formation is one of the major open questions in cloud and climate research. The conditions that favor drop freezing are - in a simplified summary: cold temperatures, high relative humidities and a 'good freezing ability'. For more details on the specific conditions see e.g. Kanji et al. (2017) and references therein.

page 3, line 13: Hallett-Mossop could be briefly explained (at least rime splintering could be mentioned).

Thank you, we have inserted rime splintering:

the Hallett-Mossop process (also called rime splintering, Hallett and Mossop, 1974)

page 3, line 13 and line 28: Frozen droplet shattering is another process which can produce secondary ice (Mason and Maybank, 1960, Leisner et al. 2014, Lawson et al. 2015).

We've added the droplet freezing here and the Leisner et al. reference on page 15, line 6f, as it refers to thunderstorms.

Hallett-Mossop process (Hallett and Mossop, 1974), droplet freezing (Lawson et al. 2015), or ice-ice collisions (Yano and Phillips, 2011). page 16, line 6: Alternatively, other ice multiplication processes (e.g. ice splintering or plasma-induced particle shattering due to lightning, see Leisner et al. 2014) might take place.

page 3, line 14/15: Why only via contact freezing? The WBF process also takes place more efficient if there are more ice crystals in the close surroundings of supercooled water droplets.

This sentence was meant to point out that under these conditions, the glaciation can even occur quickly if WBF conditions are not met. We've rephrased it therefore to:

When one of these processes has started, the remaining liquid fraction of a cloud can glaciate quickly via contact freezing, even if the conditions for the WBF process are not met.

 Contact freezing mostly refers to a collision of an aerosol particle with a supercooled droplet leading to freezing. In the case described the ice nucleus will be the ice crystal produced by secondary ice formation. An ice crystal is a perfect ice nucleus, but might be a bit confusing for some readers since ice nucleus is mostly associated with aerosol particles. Maybe it would be better to call it "collision freezing"?

Thank you for this suggestion.

the remaining liquid fraction of a cloud can glaciate quickly via freezing initiated by ice crystals colliding with supercooled water droplets.

 In case of sedimentation of ice particles it would not be a purely liquid cloud (or do the ice particles melt when falling into the cloud?)

The existence of single ice particles can almost never be excluded. Strictly speaking, this would then be a mixed-phase cloud. To account for this, we later call this cloud type "mostly liquid". On page 3, line 23 we've deleted the words 'purely liquid'.

...four types of mpt clouds are expected to occur: The first type describes clouds with many small...

 What is meant by active sensors? In-situ sensors?

Active remote sensing techniques "actively" emit radiation and measure the signal returned by the atmosphere.

Active remote sensing techniques have been used to derive liquid and...

 From which size on are they counted as ice particles?

In Taylor et al., from 50 micrometers on.

 What is the lower threshold of the asphericity measurement of small particles?

We define the asphericity threshold as the maximum size-dependent cross-polarized backscattering intensity that is caused by spherical droplets of a warm cloud (compare Figure 4 with extended caption - the threshold is shown as thin line in the bottom of the panels).

 It is not clear here what is meant by "1 Hz data".

We've added more information:

a statistical analysis of the data obtained by the NIXE-CAPS instrument with 1 Hz along the flightpath.

 How many clouds were sampled within these 38.6 hours? That would be a valuable and interesting information, especially in terms of the occurrence statistic in the end.

We agree that this would be an interesting information, however, it is very difficult to define "one cloud" in a satisfying way (compare Korolev et al., 2017). Layer structures, patchy cloud covers, and the occasional multiple sampling of the same cloud evolving with time would lead to a confusing number of categories. We therefore prefer to sum up our findings only in terms of temperature bins, which will hopefully provide a representative overview once the database is large enough.

 Why was the flying speed of HALO so high? Is that related to the aircraft itself or due to meteorological conditions?

HALO's speed only depends on the ambient air density (i.e. the altitude), it was not related to strong winds. Around 0 degC its speed was as low as 130 m/s, however, the higher the altitude, the faster HALO is flying. In ACRIDICON flight AC13, for example, HALO reached 220 m/s at -35 deg C.

 Is the concentration limitation at high aircraft speed a problem? What are typical concentrations? How many particles are missed? That would be an interesting information to add.

In thin clouds such as cirrus clouds, the concentration limitation can be an issue. The clouds presented here, however, have concentrations that result in multiple particles within the instruments' sampling volumes per second, even at HALO's highest speed. We therefore do not assume that we are missing particles. The typical concentrations are shown in Figure 6.

 What is the consequence of the possibility of near spherical ice crystals? Is that accounted for in the uncertainty estimate and how?

Near spherical ice crystals can lead to an underestimation of the glaciation degree, if aspherical fractions are used as criterion. We account for this by defining all measured aspherical fractions as "minimum aspherical fractions" - the actual ice fraction might be higher (p.11, l.20). However, we do not assume that this error often influences the aspherical fractions. According to Järvinen et al., spherical ice occurs under specific condition, e.g. in the sublimation zone of cumulonimbus anvils. These areas were not probed in the dataset shown here.

 What is expressed by the shadow intensity?

The shadow intensity shows if for the respective sensor diode, the incident laser light is fully blocked (full shadow) due to the passing particle, or if the particle is only causing a partial shadow on the diode (partial shadows are further divided into a shading of up to 35% or 35-65%).

Shadow image pixels are defined by shadow intensities of 100%-65%, 65-35%, and 35%-0% of the incident light.

page 7, line 32: Why is the smaller particle fraction measured by the CAS? Is that more sensitive in this size range compared to the CIP? How well do the number concentrations of both instruments overlap?

Particles smaller than about 20 micrometers can not be measured with the CIP technology. Refraction and diffraction might cause image artifacts and result in mis-sizing. Therefore, instead of shadow images, the scattering intensities are used for particles smaller than 50 micrometers. In the overlapping size range between 20 and 50 micrometers, the agreement is generally good for our instruments, although it depends on the respective cloud conditions. At very high cloud particle concentrations, the CIP might underestimate the particle concentrations due to coincidence. At very low concentrations, the CAS' small sampling area might lead to an underrepresentation of the particle concentration in the CAS for particles larger than 20 micrometers. (The PhD thesis of Anja Costa provides more detailed information on these errors.)

page 8, line 8: Is the limitation to 300 particles per second reached often?

For the fast-flying HALO aircraft where the sampling volume per second is large, it is reached often within clouds with concentrations larger than 10 cm$^{-3}$. The sample size of 300 particles is, however, also in these cases a good statistical basis for the asphericity analysis. The CAS' number concentrations, on the other hand, are not influenced by this particle limit: for the particle count and sizing, the 'histogram' dataset is used, which contains less detailed information per particle but has no count limit per second.

page 8, line 11: What is an inter-arrival time correction? It would help the reader to understand the results if you add a one-sentence explanation here (it appears again in section 3.5.2).

We've added a sentence:

...additionally to the instrument modifications described above. This correction rejects particles if their IATs are significantly shorter than those of majority of ice crystals, as these short IATs might result from shattering.

page 9, line 30: You could specify which lines the temperature groups refer to.

We have added:

Again, the two temperature groups are seen as for the Type 1 clouds (Figure 7), with a clear accumulation of mass at larger particle sizes for temperatures below 247.5 K.

page 9, line 32: This might be only true for the clouds where immersion freezing triggers the formation of ice crystals. That can be very different, especially for the convective tropical clouds, where ice crystals can sediment
5 from colder regions of the cloud.

Our sentence does not take into account that the ice crystals present in Type 2 clouds might result from other mechanisms. We therefore add to the last part of the sentence:

all liquid droplets evaporate leaving only the ice crystals that have already formed, e.g. via immersion freezing or ice seeding.

10 page 10, line 4: The cloud particles are droplets or ice particles or both here?

They are most likely ice particles. To say that with certainty, we have to include the asphericity analysis, which we do in section 3.3 (compare also Figure 11, right panel).

page 10, line 15: In DeMott et al. 2010 they do not limit the parameterization to 3 mum. The aerosol fraction estimated with NIXE-CAPS might therefore be underestimated (also because the lower threshold is at 0.6 mum
15 instead of 0.5 mum). It would be nice if you could add some uncertainty estimation concerning this or some argumentation for your approach.

We've added a sentence:

NIXE-CAPS records particles larger than 0.6 um; the fraction from 0.6 um to 3 um is used as 'aerosol fraction'. Due to the slightly smaller range of our aerosol measurements, the INP numbers might be underestimated.
20 However, we believe that this uncertainty is small in comparison to the DeMott parametrization itself, since (i) the difference at the lower sizes is only 0.1 micrometer and (ii) aerosol particles larger than 3 micrometers contribute only very little to the concentration of particles larger than 0.5 micrometer (see e.g. Lachlan-Cope, 2016). Their purpose is to show differences found between the measurement campaigns and temperature ranges.

page 10, line 15: The "aerosol data" used here are all particles in this size range so also all kind of hydrometeors?
25 How good does that reflect the actual aerosol concentration? Are there some aerosol measurements for one of the four campaigns, where the approach could be evaluated against?

We can not differentiate between aerosol and very small hydrometeors below 3 micrometers. However, such small hydrometeors will rapidly grow to large sizes, so we think that it is reasonable to assume that particles in

the size range 0.6 to 3 micrometer represent aerosol particles. Unfortunately, there are no measurements of solely aerosol particles available.

page 10, line 19: How often is the maximum of N-INP reached and in which cases?

Figure 10 shows the frequency of occurrence of INP concentrations per temperature bin. It can be seen that the high INP concentrations were very rare. INP concentrations of more than $1\,\text{cm}^{-3}$ were reached in less than 1% of all cases per temperature bin.

page 10, line 33: "...formed around INP." might sound a bit mis leading, you could replace it by "...initial ice crystals have likely formed by immersion freezing" or "...and that the formation of the initial ice crystals has been likely initiated by INP immersed in the cloud droplet".

Thank you for this suggestion.

...and that the formation of the initial ice crystals has been likely initiated by INP immersed in the cloud droplet.

page 11, line 1: Could that problem be solved with a size-dependent ice nucleation parameterization? That would be a very interesting aspect to look at (if not in this paper then maybe discuss this shortly here).

In the cirrus cloud research supported by NIXE-CAPS data, size-dependent ice nucleation schemes were able to fill the gap below the NIXE-CAPS measurement limit (Luebke et al. 2016). Therefore it would be a promising approach to simulate the measured clouds to investigate the low concentration range. The respective model would need to include not only ice nucleation to assess the number of small ice crystals, but also sedimentation, and collection mechanisms such as aggregation and graupel formation. We are, however, currently not involved in any project with respect to mixed-phase cloud modelling and unfortunately do not have the expertise ourselves. Thus we would like to pass on your suggestion:

Detailed microphysical cloud simulations might help to further investigate this concentration range.

page 12, line 22: Were the campaigns mostly located over open water or ice? That could explain missing marine ice nuclei. However, the freezing efficiency of these ice nuclei is rather low compared to other aerosol species. Thus even if present it could be that the clouds might not freeze at low temperatures. You could use the parameterization given in Wilson et al. 2015 and the estimated aerosol concentration to check for a few cases how high the freezing probability would be.

RACEPAC and VERDI took place during spring, when the sea ice was disappearing. During both campaigns a polynya over the Beaufort Sea was present and provided a substantial area of open water.

We are not sure if we understand the second part of the comment. Figure 2 in Wilson et al. shows that the investigated marine INP are very efficient at temperatures warmer than -20degC. So if marine aerosol is available, there should be efficient INP present.

page 12, line 30: Can you give a range of values for "very low updrafts".
We added:
...of 0.1 m/s and lower while mostly fluctuating around zero, ...

page 13, line 12: The WBF depends only on the presence of INP in "classic" stratocumulus mixed-phase cloud cases. In convective tropical clouds it could also be triggered by sedimenting ice from colder parts of the cloud.
We added:
...of INP (or seed ice from higher cloud layers)...

page 13, line 13: It is actually difficult to see a strong difference in Fig. 13 if it is plotted like this. In the current representation it looks actually like the quantities are higher in the Arctic?
That is correct. The higher overall concentrations in the Arctic at warm temperatures might be due to the lower altitude in which these temperatures occur (compare Figure 14). We speculate, however, that the Arctic aerosol might contain less efficient INPs, i.e. less marine INPs, than the mid-latitude aerosol. We've added this thought to the discussion in section 3.5.1:
As the estimated INP concentration based on aerosol measurements do not show clear conditions in the Arctic, a possible explanation[...] This might explain the lack of ice crystals, even though - possibly due to the low altitude of those warm layers - the overall aerosol concentration is comparable to the mid-latitudes.

page 13, line 17: Is that already clearly proven that biological particles occur less frequently compared to mineral dust? That might be different in the Arctic or over the Southern Ocean.
Our formulation was too strong. We have therefore rephrased this part to:
At temperatures below about -20 °C, for example, efficient mineral dust INP might initiate the freezing process, while at warmer temperatures less frequently occurring biological particles most likely act as INP (Augustin-Bauditz et al., 2014; Kanji et al., 2017).

page 13, line 25: Do you have a hypothesis why secondary ice clouds appear more often in the mid-latitudes compared to the Arctic?

We've added:

...which might reflect the increased availability of initial ice.

page 14, line 5: You could add another sentence as an explanation why secondary ice is more likely to form in turbulent environments.

We've added:

... which is consistent with the idea that the cloud particles need to collide during the rime-splintering process.

page 14, line 9-11: From Fig. 12 I can not recognize anything mentioned in the text referring to the tropical vertical velocity, neither that they reach from -10 to -15 m s^-1, nor that velocities of 0.5 to 1 m s^-1 are reached in more than 10that the distribution if wider compared to the other cases. Maybe the plotting scale is wrong or the representation of the data inappropriate?

Due to safety restrictions, the extreme vertical velocities were measured so rarely that we've decided not to include them in the plot to maintain comparability with the other panels. For clarity, we've now added:

The records... (not shown here).

page 14, line 15: It could point to biological INP, it could also point to strong sedimentation seeding the lower cloud levels.

Thank you for this valid point. We've included this now:

....also for warmer temperatures. This might be a consequence of sedimenting ice, or it might indicate...

page 14, line 17: The focus of the DeMott et al. 2010 parameterization is not on dust. For the regions investigated all aerosol species are represented, which have an ice efficiency that leads to a frozen fraction larger than the detection limit of the instrument.

We have removed the respective remark.

page 16, line 3: Would it not be possible that these small ice crystals come from secondary ice formation as well?

We can not exclude this, however, ice multiplication is usually associated with very high particle number concentrations, which we do not observe here. In contrast, the observed number concentrations match the expected INP concentrations, which seems indicate immersion freezing rather than ice multiplication.

page 16, line 11: That (Wilson et al. 2015) might not be the best reference here. It would be better to cite ice nucleation field studies from the Arctic or the BACCHUS database, which was used in Wilson et al. 2015.

We have included a remark that the Wilson study is based on field measurements.

Wilson et al. (2015), which is based on field measurements.

page 16, line 22-25: It would be nice to have this aspect a bit more detailed, maybe adding the Pruppacher et al. estimates in Fig. 15 or have a separate figure for a comparison.

Thank you for this suggestion. We have added a Figure (16) adapted from Pruppacher et al., which shows our findings in comparison to the references therein:

[Figure]

**Figure 1.** The colored lines show the fraction of clouds containing no ice found during our campaigns compared to the findings reported in Pruppacher et al. (1998).

Figure 1: The homogeneous freezing and ice multiplication cloud should be at the same location on the x-axis-both are fully glaciated.

'Homogeneous freezing' was shifted to the right. The lower part of 'ice multiplication' indicates the lower ice fraction at the beginning of the process.

Figure!

Figure 1: Coexistence is not really a path, it is clear from the x-axis that in this region there is coexistence. Maybe the different RHw areas could be colored in the background to also account for this aspect?

We've changed 'coexitence' to 'maintaining coexistence' to emphasize the dynamical aspect.

See Figure in paper.

5      Figure 1: Why is there an arrow pointing from Coexistence to homogeneous freezing clouds?

If the coexistence is maintained long enough, for example in tropical convective clouds, homogeneous freezing can take place as soon as the respective temperature threshold is reached. (In theory, homogeneous freezing could also take place in a formerly fully liquid cloud which does not contain any ice before it reaches the homogeneous freezing threshold. We are, however, unaware that such a transition has ever been observed in a natural cloud.)

10      Figure 1: Is there a reason for the WBF process to be located at -17degC?

The temperature inscriptions on the y-axis are misleading. We have replaced the middle values by the word 'cooling'.

See Figure in paper.

Figure 1: Especially in convective clouds instead of initial freezing there could be an interaction between the
15    homogeneous freezing and the mpc cloud by sedimentation of ice crystals and thus a seeding of lower cloud regions.

We've added 'ice seeding' to 'initial freezing'.

See Figure in paper.

Figure 1: The different cloud types could be added in colors to the sketch.
20    Thank you for this suggestion.

See Figure in paper.

Figure 2: Where in the figure is (approx.) 235 K? Add a line to the corresponding altitude.

The reference to homogeneous freezing is not necessary here. We have therefore removed the text '<235 K all ice'.

25    See Figure in paper.

Figure 2: Where exactly does the text < 235 K all ice refer to? Should not the drop growth curve then end at 235 K?

See comment above.

Figure 4: What is the blue line in the plots?

We've added:

The horizontal line in the bottom of the panels shows the signal intensity in the S-pol detector which must be exceeded for a particle to be detected as 'aspherical'.

Figure 5: What does the color coding in the upper panel stand for? dN/dlogDp?

Yes. We've added:

Upper panel: ...of the VERDI campaign (color code: dN/dlogDp).

Figure 5: The labels of the color bar are too small.

Thank you for pointing this out, we have corrected it.

See Figure in paper.

- Figure 7 and 8: The second line is not at 20 mum.

Thank you for pointing this out.

See Figure in paper.

Figure 7 and 8: The lower panel is not explained in the caption or text. What is shown here? What does that show in addition to the size distributions? Do the stripes correspond to different diameters?

We've added a more detailed explanation to the figure inscription and a scale to the CIP image stripes.

Example of CIP images (background picture). The stripes represent a series of CIP shadow images, depicting the particles that have passed subsequently through the detector. Foreground: Average particle...

Figure 9: The number of INP plotted nearly follows the constant aerosol concentration lines- does that only look like it or is the concentration not so variable with height/temperature?

At temperatures below 252 K, note that the highest frequencies of occurrence are 10-20%; the distribution is thus relatively wide-spread. At the warmer temperatures, you are right. In this temperature range, very high aerosol numbers have to be present to produce these low INP concentrations. The limited aerosol particle size range that can be observed by NIXE-CAS might contribute to the narrow distribution of INP concentrations found here.

5     Figure 9: The color coding could be mentioned in the caption.

We've added a second sentence:

INP number concentrations ($N_{INP}$) vs. temperature for all measurement campaigns, color coeded by their frequency of occurrence. $N_{INP}$ is estimated from NIXE-CAPS measurements of aerosol concentrations.

Figure 9: Why is the temperature range limited here compared to Fig. 13?

10     Figure 13 shows an unnecessary large temperature range. This we have corrected.

See Figure 13.

Figure 11: The Arctic campaigns look rather different compared to the other locations, why is that? Maybe also discuss that in the text with a reference to this figure.

In this plot, observations at the same temperature and with the same AF are stacked. The Arctic campaigns were

15     plotted last: since they had the smallest temperature range, they would otherwise be hidden. Therefore, this plot shows them prominently. The other campaigns, however, have shown the same cloud types at these temperatures.

Figure 11: Why are there so few data points for the VERDI campaign?

As mentioned in section 2.1, the Arctic campaigns actually contributed a large cloud data set. Due to the restricted temperature range probed during VERDI, however, the observations were not wide-spread.

20     Figure 12: Why is the vertical velocity distribution from RACEPAC not added?

At the point of the publication, no quality-checked observational data were available.

Figure 13: Is the limitation of the data points in the Arctic case due to flight altitude?

Yes. During VERDI an unpressurized aircraft was opperated. Therefore, only clouds in altitudes below 3500 m could be sampled. Due to the limited altitude range also the temperatur range is limited.

Figure 14: Is there not also a difference due to different flight altitudes?

The altitude range probed in the respective campaign differs, as it can be read from the x-axis. However, it is also clearly visible that at the same altitude (e.g. 2000 m), the temperature in the Arctic was lower than at mid-latitudes and much lower than in the tropics. These differences are important to keep in mind when comparing the other plots of this study, which rely on temperature as comparison criterion.

Table 1: Particles can not be glaciated, wording needs to be adapted.

'glaciated' was changed to 'ice crystals'.

See table 1

Table 2: What is the difference between a "Stratocumulus" and a "Stratocumulus in mixed-phase T regime"? Is there a certain temperature threshold (even below 0degC) assumed (row 5-7)? The cloud in row 1 is also likely to be "Stratocumulus in mixedphase T regime"? Or why is it a "Warm cloud"?

Thank you for pointing this out. The third column makes the remark 'in the mpt regime' unnecessary. It was therefore removed. With respect to warm clouds: The table gives a list, i.e. during the first flight, warm clouds were observed, mixed clouds and cirrus clouds. We have decided to give a complete overview of the respective campaign and therefore included also those descriptions which are not assessed in this paper (such as warm clouds, contrail cirrus, albedo flights...). This might help to encourage further studies investigating the cases presented here.

See table

Table 4: Remove 11.05. and 13.05. or are the measurements done at these days used within this paper?

As mentioned before, we would like to include these days nevertheless, as measurement data of other campaign instruments (e.g. remote sensing) are available for further studies.

Table 5: Remove 21.09., 27.09. and 30.09. or are the measurements used within this paper?

See above.

Small remarks,typos: - page 1 line 3: Space missing between number and unit (to be consistent with the rest of the paper). - page 1, line 7: Replace associated with by : . - page 1, line 13: You might also want to specify the temperature range for the tropics. - page 1, line 15: The second "to" is too much. - page 2, line 21: Delete "nature

of" (it is not the nature of the properties...). - page 2, line 31: "with modification" is redundant (already written "adapted from"). - page 3, line 29: Verb missing.

Thank you, we have corrected these parts.

Text changes - see paper

5      - page 4, line 16/17: The clouds after the WBF process could eventually be named as 'glaciated clouds' (also this can be a bit ambiguous) or 'WBF glaciated clouds'. Or maybe it would be good to introduce the names here that are later on used, i.e. page 11 and 12.

We've added:

... and 'large ice' clouds after the WBF process

10     - page 4, line 22: Add "(cloud spectrometer)". - page 4, line 18-21: Sentence is too long and difficult to read. - page 6 line 8: It would be nice to add a reference where NIXE-CAS-DPOL and NIXECIPg are explained later on in the text. - page 6, line 13: One bracket too much. - page 8, line 5: What does the "With these" refer to? - page 9, line 10: One bracket too much. - page 11, line 21: Shift this sentence to the beginning of this section. - page 12, line 27: "reflected" instead of "reflect". - page 12, line 24: The reference Augustin-Bauditz et al., 2014 does not

15     fit in the context. - page 13, line 10: Delete "and". - page 13, line 22: replace the "-" with something equivalent, it could look like -253 K. - page 13, line 22: There is a ":" too much. - page 14, line 27: Switch bracket and "clouds". - page 14, line 34: "darkblue" instead of "blue". - page 15, line 27: "On the contrary" does not make sense since the statement is further supported? - Caption Fig. 1: . missing at the end. - Caption Fig. 2: Add "z" after altitude. - Caption Fig. 2: Remove the bracket WBF..., that is not written in the figure. - Figure 4: There are some black dots

20     around the axis labels at the right panel of the figure. - Figure 6: Delete the 1 in the unit-brackets. - Figure 7 and 8: The blue line is not thick and not so easy to differentiate from the others. - Figure 13: RACEPAC is mentioned in the caption but not in the title of the figure. - Figure 15: The fonts are quite small. - Table 5: To be consistent remove "profiling". - Table 5: Write out "Cb".

Thank you, we have corrected these parts.

25     Text changes: see paper

**References**

Augustin-Bauditz, S., Wex, H., Kanter, S., Ebert, M., Niedermeier, D., Stolz, F., Prager, A., and Stratmann, F.: The immersion mode ice nucleation behavior of mineral dusts: A comparison of different pure and surface modified dusts, Geophysical Research Letters, 41, 7375–7382, 2014.

5   Curry, J. A., Schramm, J. L., Rossow, W. B., and Randall, D.: Overview of Arctic Cloud and Radiation Characteristics, Journal of Climate, 9, 1731–1764, doi:10.1175/1520-0442(1996)009<1731:OOACAR>2.0.CO;2, https://doi.org/10.1175/1520-0442(1996)009<1731:OOACAR>2.0.CO;2, 1996.

Kanji, Z., Ladino, L. A., Wex, H., Boose, Y., Burkert-Kohn, M., Cziczo, D. J., and Krämer, M.: Ice Formation and Evolution in Clouds and Precipitation: Measurement and Modeling Challenges, Chapter 1: Overview of Ice Nucleating Particles,
10   Meteorological Monograph, 2017.

Korolev, A., McFarquhar, G., Field, P., Franklin, C., Lawson, P., Wang, Z., Williams, E., Abel, S., Axisa, D., Borrmann, S., et al.: Ice Formation and Evolution in Clouds and Precipitation: Measurement and Modeling Challenges. Chapter 5: Mixed-phase clouds: progress and challenges., Meteorological Monographs, 58, 2017.

Pruppacher, H. R., Klett, J. D., and Wang, P. K.: Microphysics of clouds and precipitation, Taylor & Francis, 1998.

15   Shupe, M. D. and Intrieri, J. M.: Cloud Radiative Forcing of the Arctic Surface: The Influence of Cloud Properties, Surface Albedo, and Solar Zenith Angle, Journal of Climate, 17, 616–628, doi:10.1175/1520-0442(2004)017<0616:CRFOTA>2.0.CO;2, https://doi.org/10.1175/1520-0442(2004)017<0616:CRFOTA>2.0.CO;2, 2004.

---

## Author Response (AR2)

**Classification of Arctic, Mid-Latitude and Tropical Clouds in the Mixed-Phase Temperature Regime**

Anja Costa[1], Jessica Meyer[1,2], Armin Afchine[1], Anna Luebke[1,3], Gebhard Günther[1], James R. Dorsey[4], Martin W. Gallagher[4], Andre Ehrlich[5], Manfred Wendisch[5], Darrel Baumgardner[6], Heike Wex[7], and Martina Krämer[1]

[1]Forschungszentrum Jülich GmbH, Jülich, Germany
[2]now at: Bundesanstalt für Arbeitsschutz und Arbeitsmedizin, Dortmund, Germany
[3]now at: Max Planck Institute for Meteorology, Atmosphere in the Earth System Department, Hamburg, Germany
[4]Centre for Atmospheric Science, University of Manchester, UK
[5]Leipziger Institut für Meteorologie, Universität Leipzig, Germany
[6]DMT, Boulder/Colorado, USA
[7]Leibniz Institute for Tropospheric Research, Leipzig, Germany

*Correspondence to:* Martina Krämer (m.kraemer@fz-juelich.de)

Dear Corinna Hoose,

thank you for your suggestions, we have implemented almost all of them. To some of your comments on the figures we would like to answer:

With regard to Figure 15, we would like to keep the detailed caption to enable the reader to treat this figure as a standalone summary of our observations.

With respect to the remark on Table 1 and the table in Figure 15: Table 1 summarizes assumptions on the clouds which we expect to find. As a part of the introduction, we feel that it should not include our methods. Fig. 15, in contrast, shows our measurement results (aspherical fractions, large particle images, quantitative size distributions). Therefore, we do not think that this is an exact reproduction of the introductory table. Also, we think that adding the PSD sorting algorithms to the table in Fig. 15 will make it too complicated. However, we have added a reference to the PSD section.

Thank you again for your quick and smooth handling our manuscript!

Best wishes,

Anja Costa and coauthors